# B cell receptor-induced IL-10 production from neonatal mouse CD19⁺CD43⁻ cells depends on STAT5-mediated IL-6 secretion

Jiro Sakai[1], Jiyeon Yang[1], Chao-Kai Chou[2], Wells W Wu[2], Mustafa Akkoyunlu[1]*

[1]Laboratory of Bacterial Polysaccharides, Division of Bacterial Parasitic and Allergenic Products, Center for Biologics Evaluation and Research, The US Food and Drug Administration, Silver Spring, United States; [2]Facility for Biotechnology Resources, Center for Biologics Evaluation and Research, United States Food and Drug Administration, Silver Spring, United States

**Abstract** Newborns are unable to reach the adult-level humoral immune response partly due to the potent immunoregulatory role of IL-10. Increased IL-10 production by neonatal B cells has been attributed to the larger population of IL-10-producing CD43⁺ B-1 cells in neonates. Here, we show that neonatal mouse CD43⁻ non-B-1 cells also produce substantial amounts of IL-10 following B cell antigen receptor (BCR) activation. In neonatal mouse CD43⁻ non-B-1 cells, BCR engagement activated STAT5 under the control of phosphorylated forms of signaling molecules Syk, Btk, PKC, FAK, and Rac1. Neonatal STAT5 activation led to IL-6 production, which in turn was responsible for IL-10 production in an autocrine/paracrine fashion through the activation of STAT3. In addition to the increased IL-6 production in response to BCR stimulation, elevated expression of IL-6Rα expression in neonatal B cells rendered them highly susceptible to IL-6-mediated STAT3 phosphorylation and IL-10 production. Finally, IL-10 secreted from neonatal mouse CD43⁻ non-B-1 cells was sufficient to inhibit TNF-α secretion by macrophages. Our results unveil a distinct mechanism of IL-6-dependent IL-10 production in BCR-stimulated neonatal CD19⁺CD43⁻ B cells.

*For correspondence:
Mustafa.Akkoyunlu@fda.hhs.gov

Competing interest: The authors declare that no competing interests exist.

## Editor's evaluation

In this study, authors demonstrate that neonatal mice produce more CD43- B cell-derived IL-10 following anti-BCR stimulation than adult mice. This is due to a autocrine mechanisms where by anti-BCR stimulation leads to pSTAT5 up regulation, production of IL-6 which then enhances IL-10 production via pSTAT3. These are interesting results for the regulatory B cell field, demonstrating that signaling is different in adult vs neonatal B cells and in particular for researchers studying the mechanisms underpinning the enhanced susceptibility to infection.

## Introduction

Infants and neonates demonstrate high susceptibility to infection, leading to 40% of the annual death of approximately 7 million children under the age of 5 years worldwide (*Bhutta and Black, 2013*). Fetomaternal immune tolerance is essential to suppress rejection during pregnancy, and disruption of the fetomaternal tolerance can result in preterm labor (*Gomez-Lopez et al., 2014*). During the perinatal period, newborns transition from their dependence on maternal immunity to their own immune system (*Simon et al., 2015*). Rapid exposure to environmental assaults, such as microbes,

after birth renders neonates susceptible to infections (*MacGillivray and Kollmann, 2014*; *Kollmann et al., 2017*). Although vaccination has been successful in curbing early age infection rates, infants are still vulnerable during to first year of life, especially because most pediatric vaccines need to be administered four times during the first 15 months of age in order to elicit adult like protective immunity against infections (*Ehreth, 2003*; *Jacobson, 2020*). The exact reasons for the suboptimal vaccine responses after birth are not sufficiently delineated and this knowledge gap is an obstacle in improving pediatric vaccines (*Kollmann et al., 2017*). The protective host response to most vaccines is initiated by the recognition of vaccine antigen via the B cell receptor (BCR) (*Akkaya et al., 2020*), followed by the generation of germinal center (GC) response that involve follicular helper T (Tfh) cells and GC B cells in secondary lymphoid organs (*Vinuesa et al., 2016*). We and others have shown that the unique properties of impaired Tfh response contribute to the weak immune response to vaccines in neonatal mice (*Mastelic et al., 2012*; *Yang et al., 2018*). In addition, several types of immune suppressive cells contribute to fetomaternal immune tolerance and to human and murine neonatal suboptimal immunity via the potent anti-inflammatory cytokine interleukin (IL)–10 (*Kollmann et al., 2017*; *Basha et al., 2014*).

Initially described as a function of activated CD4$^+$ T helper (Th) 2 cells to inhibit cytokine production by Th1 cells (*Fiorentino et al., 1989*), IL-10 has subsequently been shown to be produced by different cell types including dendritic cells, macrophages, neutrophils, mast cells, natural killer cells, T cells, and B cells (*Moore et al., 2001*; *Ouyang and O'Garra, 2019*). The main activity of IL-10 is the inhibition of inflammatory responses (e.g., pro-inflammatory cytokine and chemokine synthesis, nitric oxide production, and antigen presentation) by both the innate and the adaptive immune cells (*Moore et al., 2001*; *Ouyang and O'Garra, 2019*). IL-10 expression is controlled by various transcription factors including nuclear factor-κB (NF-κB), signal transducer and activator of transcription 3 (STAT3), and GATA binding protein 3 (GATA3) depending on the upstream signaling pathways and cell types (*Saraiva and O'Garra, 2010*; *Hedrich and Bream, 2010*). In human and neonatal B cells, IL-10 production has been shown to be mediated by a range of signaling, including BCR engagement (*Alhakeem et al., 2015*), CD40 ligand (*Burdin et al., 1997*), TLR agonists (*Lampropoulou et al., 2008*), IL-1β, IL-6 (*Rosser et al., 2014*) and type I interferons (*Zhang et al., 2007*). In this study, we focused on the differences between adult and neonatal BCR induced IL-10 production to begin understanding the contribution of BCR mediated IL-10 production in weak vaccine responses in neonates.

B cells can be divided into two subsets: CD43$^+$ B-1 cells and CD43$^-$ non B-1 cells which include follicular B-2 cells, marginal zone B cells and IL-10 producing regulatory B cells (Bregs) (*Baumgarth, 2011*). B-1 cells are further subdivided into innate-like B-1a (CD5$^+$) cells and B-1b (CD5$^-$) cells. After emerging from the liver in the very early stage of life, B-1 cells have been suggested to be maintained by a 'self-renewal' process, whereas B-2 cells are continuously produced from progenitors in the bone marrow after birth in mice and humans (*Baumgarth, 2011*; *Montecino-Rodriguez and Dorshkind, 2012*). Accordingly, murine B-1 cells, which dominate the B cell population in neonatal spleen (30% of IgM$^+$ splenic B cells), are gradually exceeded by B-2 cells with age and account for approximately 2% in adult splenic B cells (*Hayakawa et al., 1983*; *Montecino-Rodriguez and Dorshkind, 2011*). Among the B cells, B-1 cells are the main producers of IL-10 (*O'Garra et al., 1992*; *Zhang, 2013*) and since B-1 cells account for 1/3 of neonatal mouse spleen, they are considered as the main source of IL-10 in neonatal spleen. Both mouse neonatal splenic and human cord blood Breg cells are shown to manifest suppressive activity (*Zhang, 2013*; *Sarvaria et al., 2016*; *Esteve-Solé et al., 2017*).

Here, we demonstrated that there is a sizable neonatal mouse splenic CD19$^+$CD43$^-$ non-B-1 cell population with potent IL-10 producing capacity in response to BCR stimulation. Upon BCR stimulation, neonatal CD19$^+$CD43$^-$ cells, but not their adult counterparts, produced IL-6 downstream of activated STAT5. In an autocrine and paracrine fashion, the secreted IL-6 triggered the induction of IL-10 production from CD19$^+$CD43$^-$ cells. Further, neonatal CD19$^+$CD43$^-$ cells suppressed inflammatory cytokine production by macrophages in an IL-10-dependent manner.

## Results

### Neonatal spleen contains a CD43⁻ cell population with substantial IL-10 production in response to BCR stimulation

The IL-10-producing splenic B cell population includes several different subsets characterized by surface markers (*Rosser and Mauri, 2015*). Among these subsets, the CD5$^+$ B-1 B cells comprise a small portion of adult spleen, but represent a higher percentage of neonatal splenic B cells (*Hayakawa et al., 1983*). Accordingly, the higher representation of IL-10$^+$ B cells in neonatal spleen following a variety of stimuli is thought to be due to higher percentage of CD5$^+$ B-1 cells in neonatal spleen (*Sun et al., 2005*; *Yanaba et al., 2009*). B-1 cells are identified by CD43 expression, and CD5 is used to distinguish between B-1a (CD43$^+$CD5$^+$) and B-1b (CD43$^+$CD5$^-$) (*Baumgarth, 2004*; *Hardy, 2006*). In this study, we sought to characterize the generation of neonatal IL-10-producing B cells following BCR stimulation as antigen recognition by B cells is essential for the activation of B cells in response to vaccines (*Akkaya et al., 2020*). First, we purified CD19$^+$ cells from splenocytes (*Figure 1—figure supplement 1*). Intracellular IL-10 staining confirmed the increased emergence of CD19$^+$IL-10$^+$ B cells after stimulation of purified neonatal splenic B cells with anti-IgM antibodies compared to adult B cells (*Figure 1A*). Also confirming previous reports (*Hayakawa et al., 1983*; *Sun et al., 2005*), neonatal spleen contained higher percentage of CD43$^+$ B-1 cells than adult spleen (*Figure 1—figure supplement 2A* and B). Further gating of purified B cells based on CD43 expression indicated that neonatal CD43$^+$ cells contained higher percentage of CD19$^+$IL-10$^+$ B cells than adult cells prior to BCR stimulation and there was a modest increase after BCR stimulation in both the age groups (*Figure 1B*). Unlike the CD43$^+$ population, BCR stimulation induced a significant increase in IL-10$^+$ cells among the neonatal CD43$^-$ population, whereas IL-10$^+$ cells remained low in adult CD43$^-$ population after BCR-stimulation (*Figure 1B*). Thus, although neonatal splenic CD19$^+$CD43$^+$ cells comprise the main IL-10$^+$ cells in neonatal spleen as reported previously, there is a substantial increase in IL-10$^+$ cell frequency and mean fluorescence intensity (MFI) among the CD19$^+$CD43$^-$ population following antigen recognition (*Figure 1B* and *Figure 1—figure supplement 3*). Since the CD19$^+$CD43$^+$ cells have been well recognized as the IL-10-producing subset in neonatal spleen, we focused on the characterization of this newly defined IL-10-producing CD19$^+$CD43$^-$ cell population in neonatal mice. We purified CD19$^+$CD43$^-$ cells from splenocytes of adult and neonatal mice. There was no statistically significant difference in the CD43 MFI between adult and neonatal mice after the purification of CD19$^+$CD43$^-$ cells (*Figure 1—figure supplement 4A* and B). Furthermore, there was no significant difference in the expression of *spn* gene coding for CD43 between purified adult and neonatal CD19$^+$CD43$^-$ cells (*Figure 1—figure supplement 4C*). These results demonstrated that splenic B cells from the two age groups had comparable CD43 level after purification. Next, we measured secreted IL-10 in the culture media of purified CD19$^+$CD43$^-$ cells after stimulating with anti-IgM antibodies. As observed in the flow cytometry analysis of gated CD19$^+$CD43$^-$ cells (*Figure 1B*), purified neonatal CD19$^+$CD43$^-$ cells also produced higher IL-10 than the adult counterparts (*Figure 1C*). The increase in IL-10 production reached statistical significance at 6-hr time point and remained high after peaking at 12 hr. IL-10 production by adult CD19$^+$CD43$^-$ cells was minimal at all time points. *Il10* mRNA measurement indicated that adult CD19$^+$CD43$^-$ cells manifested a rapid and sharp increase in *Il10* expression which was quickly shut off within 6 hr post-stimulation (*Figure 1D*), and this temporal gene expression appeared to be insufficient for IL-10 protein synthesis (*Figure 1C*). An initial sharp increase of *Il10* expression was also observed in neonatal CD19$^+$CD43$^-$ cells (*Figure 1D*). Unlike in adult cells, neonatal *Il10* expression further increased at later time points, facilitating the production of IL-10 (*Figure 1C*). These results indicate that, unlike its adult counterparts, neonatal splenic CD19$^+$CD43$^-$ cells have the propensity to produce IL-10 after BCR cross-linking.

Elevated levels of membrane IgM have been observed in human cord blood B cells compared to adult cells (*Glaesener et al., 2018*; *Macardle et al., 1997*). Since we stimulate B cells through IgM, we wanted to know if the IgM expression levels differed between adult and neonatal B cells. There was no statistically significant difference in the percentage of CD19$^+$IgM$^+$ cells between adults and neonates, although neonates exhibited significantly lower frequency of CD19$^+$IgD$^+$ B cells compared to adults (*Figure 1—figure supplement 5A, B*). When we analyzed the subsets, we found that neonatal CD19$^+$CD43$^-$ cells showed higher frequency and MFI of IgM$^{hi}$ cells compared to adult counterparts, whereas IgM levels were statistically comparable between adult and neonatal CD43$^+$ B-1

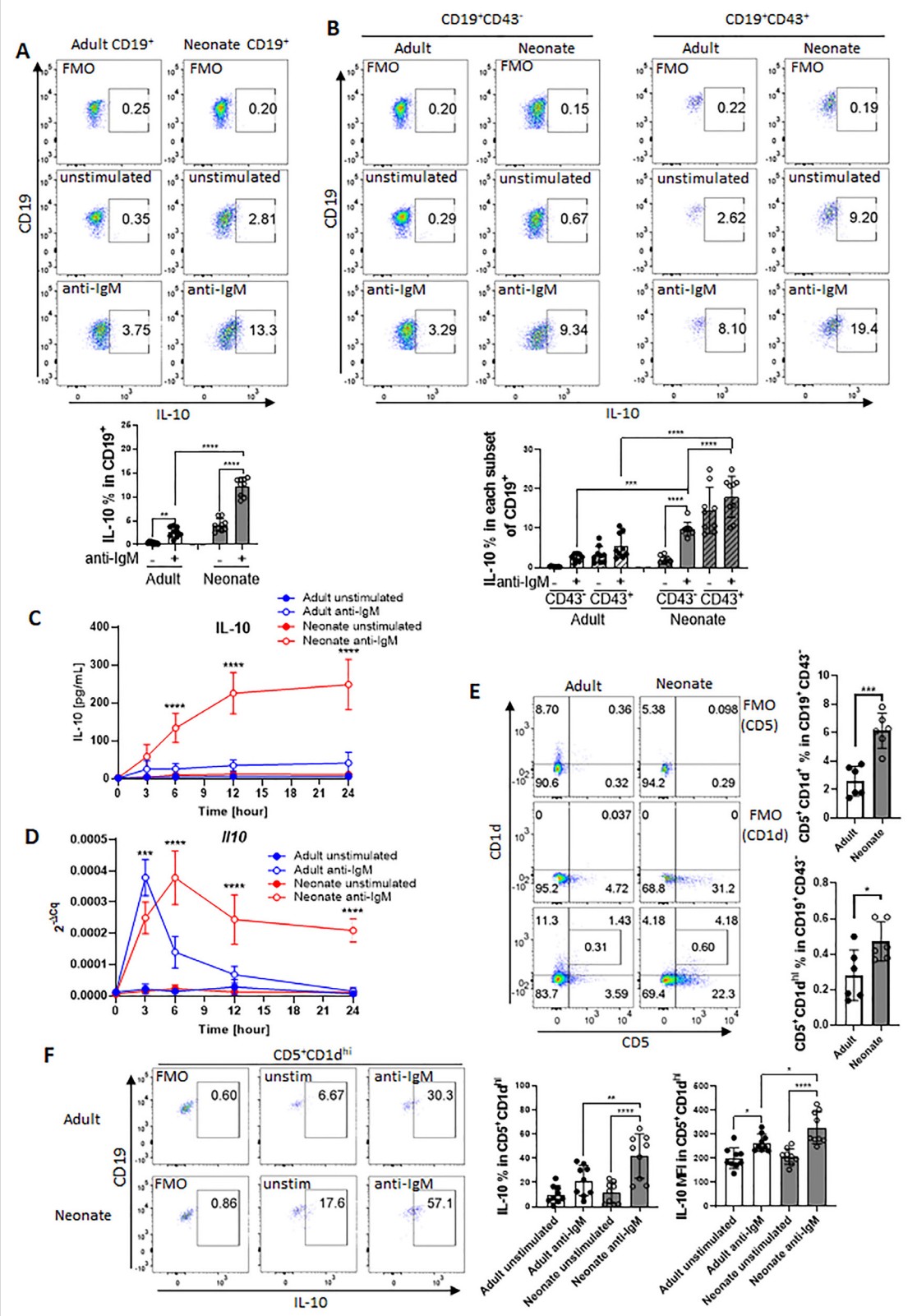

**Figure 1.** BCR-induced IL-10 production in adult and neonatal splenic B cell subsets. (**A and B**) Splenic B cells were isolated from adult and neonatal mice. Isolated cells were incubated in the absence (unstimulated) or presence of 10 μg/mL F(ab')$_2$ fragments of anti-IgM antibodies for 20 h and intracellular IL-10 production was measured by flow cytometry in gated B cell subsets (technical triplicate per experiment). Gating is dependent on fluorescence-minus-one (FMO) control. All experiments were repeated twice for reproducibility, and data are shown as the mean ± s.d. of

*Figure 1 continued on next page*

*Figure 1 continued*

three biological replicates. *P* values were calculated using one-way ANOVA with a Dunnett's multiple comparisons test (**p<0.01, ***p<0.001, and ****p<0.0001). (**C and D**) Isolated adult and neonatal CD19⁺CD43⁻ B cells were incubated at 3 million cells/mL in the absence or presence of 10 µg/mL F(ab')₂ fragments of anti-IgM antibodies for indicated duration and culture supernatant IL-10 levels (**C**) and mRNA expression (**D**) were measured in ELISA and q-PCR, respectively. Data are shown as the mean ± s.d. of three biological replicates. *P* values (Adult anti-IgM vs Neonate anti-IgM) were calculated using two-way ANOVA (***p<0.001 and ****p<0.0001). (**E**) Representative flow cytometry dot plots and the mean of CD5⁺CD1d⁺ cell-frequency (Upper) and B10 cell (CD5⁺CD1dʰⁱ)-frequency (Lower) among CD19⁺CD43⁻ B cells are plotted (technical triplicate per experiment). The experiments were repeated once for reproducibility, and data are shown as the mean ± s.d. of two biological replicates. *P* values were calculated using two-tailed Student's *t*-test (***p<0.001). (**F**) Isolated CD19⁺CD43⁻ B cells were incubated in the absence (unstim) or presence of 10 µg/mL F(ab')2 fragments of anti-IgM antibodies for 17 hr, and then analyzed for intracellular IL-10 production in flow cytometry (technical triplicate per experiment). The experiments were repeated twice for reproducibility. Representative flow cytometry dot plots of intracellular IL-10 in B10 cells are shown. The other subsets are shown in *Figure 1—figure supplement 8*. The mean ± s.d. of frequency (Left) and MFI (Right) of IL-10-expressing subsets from three biological replicates are shown. *P* values versus adult counterparts were calculated using one-way ANOVA with a Dunnett's multiple comparisons test (*p<0.05) and two-tailed Student's *t*-test (†p<0.05).

The online version of this article includes the following figure supplement(s) for figure 1:

**Figure supplement 1.** Isolation of splenic CD19⁺ B cells.

**Figure supplement 2.** Staining of splenic B cell subsets for IL-10 expression.

**Figure supplement 3.** BCR-induced IL-10 production in adult and neonatal splenic B cell subsets.

**Figure supplement 4.** Isolation of splenic CD19⁺CD43 B cells.

**Figure supplement 5.** Neonatal splenic CD19⁺CD43 cells express higher levels of IgM.

**Figure supplement 6.** CD19⁺CD43ˡᵒʷ B cells do not contribute to IL-10 production.

**Figure supplement 7.** Identification of splenic Breg subsets in adult and neonatal spleens.

**Figure supplement 8.** CD1d and CD5 expression on adult and neonatal splenic CD19⁺CD43 B cells.

**Figure supplement 9.** BCR-induced IL-10 production in each subset of adult and neonatal splenic CD19⁺CD43 B cells.

cells (*Figure 1—figure supplement 5A*). Both neonatal CD19⁺CD43⁻ cells and CD43⁺ B-1 cells exhibited lower frequency and MFI of IgDʰⁱ cells compared to adult counterparts.

## Neonatal spleen harbors higher frequency of B10 cells among CD19⁺CD43⁻ subset as compared to adult spleen

Because our experimental set up relies on the separation of CD43⁻ cells, we wanted to assess the changes in CD43 expression before and after stimulation since anti-IgM antibodies may alter the expression of CD43 (*Björck et al., 1991*). We observed that unstimulated adult CD19⁺ cells contained only a small population of CD43⁺ cells which were approximately 1/5ᵗʰ of the neonatal CD43⁺ cells (*Figure 1—figure supplement 6*). In both age groups, CD43⁺ cell population increased after BCR engagement. Next, we performed an experiment to compare the IL-10-producing CD43⁻ population in stimulated and unstimulated cells after increasingly stringent setting of the CD43 gate. Both adult and neonatal IL-10⁺ cell populations were not affected from the stringency of the gating of CD43⁻ population in unstimulated cells. In contrast, only the frequency of IL-10-producing neonatal CD43⁻ population increased with more stringent gating of CD43⁻ cells after stimulation, whereas IL-10⁺ population did not change in adult cells regardless of the stringency of CD43 gating. These results rule out the possible impact of BCR-induced changes in CD43 expression in IL-10 production from CD43⁻ population.

We next investigated the Breg cells in neonates since they have been identified as the primary B cell population producing IL-10 among CD19⁺CD43⁻ cells (*Rosser and Mauri, 2015*). We began by measuring the populations of splenic Breg subsets: transitional-2 marginal zone precursor cells (T2-MZP; CD19⁺CD21ʰⁱCD23ʰⁱCD24ʰⁱ), marginal zone B cells (MZB; CD19⁺CD21ʰⁱCD23⁻), Tim-1 B cells (CD19⁺Tim-1⁺), and IL-10-producing B cells (B10; CD19⁺CD5⁺CD1dʰⁱ) in adult and neonatal spleens. We found that the frequencies of T2-MZP (*Figure 1—figure supplement 7A*) and MZB (*Figure 1—figure supplement 7B*) were significantly lower in neonatal spleen than in adult spleen, and the Tim-1 B cell population was comparable between the two age groups (*Figure 1—figure supplement 7C*). These results excluded the possibility of these Breg subsets participating in the production of IL-10 from neonatal CD19⁺CD43⁻ cells.

Adult B10 cells with T cell suppressive properties were first described by Yanaba and colleagues as splenic CD19[+] cells expressing CD5 and CD1d[hi] (*Yanaba et al., 2008*). Our data indicated that the frequencies of CD5[+]CD1d[+] as well as CD5[+]CD1d[hi] populations were significantly more in neonates than in adults (*Figure 1E*). Assessment of changes in the expression of individual surface markers revealed that the larger population of B10 cells in neonates was attributed to the significantly higher expression of CD5 (*Figure 1—figure supplement 8A*) rather than a difference in CD1d expression (*Figure 1—figure supplement 8B*). After stimulation of CD19[+]CD43[-] cells with anti-IgM antibody, adult and neonatal CD5[+]CD1d[+] cells were the main producers of IL-10, but the neonatal CD5[+]CD-1d[+]IL-10[+] population was higher than the adult counterparts (*Figure 1—figure supplement 9A–D*). IL-10-producing population was even higher when the cells were further gated for B10 (CD5[+]CD1[hi]) population in both age groups (*Figure 1F*). Once again, the frequency as well as the MFI of IL-10-producing B10 population was more in neonatal mice then those in adult mice. Taken together, these results indicate that the larger population of IL-10-producing B10 subset is responsible for the overall IL-10 production in neonatal splenic CD19[+]CD43[-] cells in response to BCR stimulation.

## Neonatal BCRs uniquely activate STAT3 and STAT5 following BCR cross-linking

To gain insight into the underlying mechanisms of enhanced IL-10 production in BCR-stimulated neonatal CD19[+]CD43[-] cells, we performed RNA sequencing (RNA-seq) and Gene Set Enrichment Analysis (GSEA) (*Subramanian et al., 2005*) for adult and neonatal B cells following BCR engagement. There were 1341 increased and 1905 decreased genes in adult B cells compared to 716 increased and 962 decreased genes in neonatal B cells (*Figure 2—figure supplement 1A*). To identify signaling pathways uniquely activated by neonatal BCR, we compared adult and neonatal CD19[+]CD43[-] cell data using hallmark gene sets from the Molecular Signatures database (MSigDB) C5 gene ontology (GO) collection (*Liberzon et al., 2015*). Comparative analysis of gene expression in anti-IgM-stimulated B cells against unstimulated cells indicated gene sets uniquely enriched in neonatal B cells (*Figure 2A*). The top gene sets enriched in neonatal CD19[+]CD43[-] cells after anti-IgM stimulation contained cytokine receptor signaling and STAT protein-related signaling pathways (*O'Shea et al., 2013*), whereas the gene sets involved in biological processes as well as the signaling pathways controlled by mitogen-activated protein kinases (MAPKs) and Akt (*O'Shea et al., 2013*; *Buj and Aird, 2018*; *Rosini et al., 2000*; *Van Laethem et al., 2004*; *Zhuang et al., 2016*) were highly ranked in adult CD19[+]CD43[-] cells. We also conducted GSEA using the MSigDB C3 transcription factor targets (TFT) hallmark gene collection to identify transcription factors uniquely activated in neonatal B cells. This analysis revealed that target genes for STAT3, STAT5A, and STAT5B were highly expressed in anti-IgM-stimulated neonatal B cells, suggesting that these STAT proteins were activated in neonatal CD19[+]CD43[-] cells after BCR engagement (*Figure 2B*). Supporting the findings in C5 gene ontology analysis, target genes for MAPK and Akt signaling pathway-activated transcription factors such as MYC, ELK1, and AP-1 (*Zhang and Liu, 2002*; *Hoxhaj and Manning, 2020*) were not expressed in neonatal CD19[+]CD43[-] cells compared to adult cells (*Figure 2B*). To verify the signaling pathways identified by the GSEA analysis of the RNA-seq data, we subjected the BCR-stimulated neonatal and adult CD19[+]CD43[-] cells to Western blot analysis. We found that STAT3 and STAT5 were highly phosphorylated in neonatal B cells although their activation kinetics was different (*Figure 2C, D*). STAT5 phosphorylation was detected as early as 5 min and peaked at 15 min after BCR crosslinking (*Figure 2C*), whereas STAT3 phosphorylation peaked at around 4 hr post-stimulation (*Figure 2D*). Increases in STAT5 and STAT3 phosphorylations were also observed by flow cytometry at 15 min and 4 hr post-stimulation, respectively (*Figure 2—figure supplement 1B, C*). There was no significant difference in the expression of target gene sets for STAT1 between adult and neonatal B cells (*Figure 2B*). In addition, STAT1 phosphorylation was not observed after BCR cross-linking in either adult or neonatal B cells, while STAT1 was phosphorylated in B cells stimulated with recombinant IL-21 (*Figure 2C, D* and *Figure 2—figure supplement 2*), suggesting that STAT1 was not activated by BCR and that the marker genes were expressed independently of STAT1. These results suggest no role for STAT1 in BCR-induced IL-10 production. Confirming the GSEA analysis data, Akt, p38, JNK and ERK were phosphorylated in adult B cells, but not in neonatal cells (*Figure 2—figure supplement 3*). Collectively, RNA-seq and western blot analyses revealed important differences between neonatal and adult BCR signaling pathways.

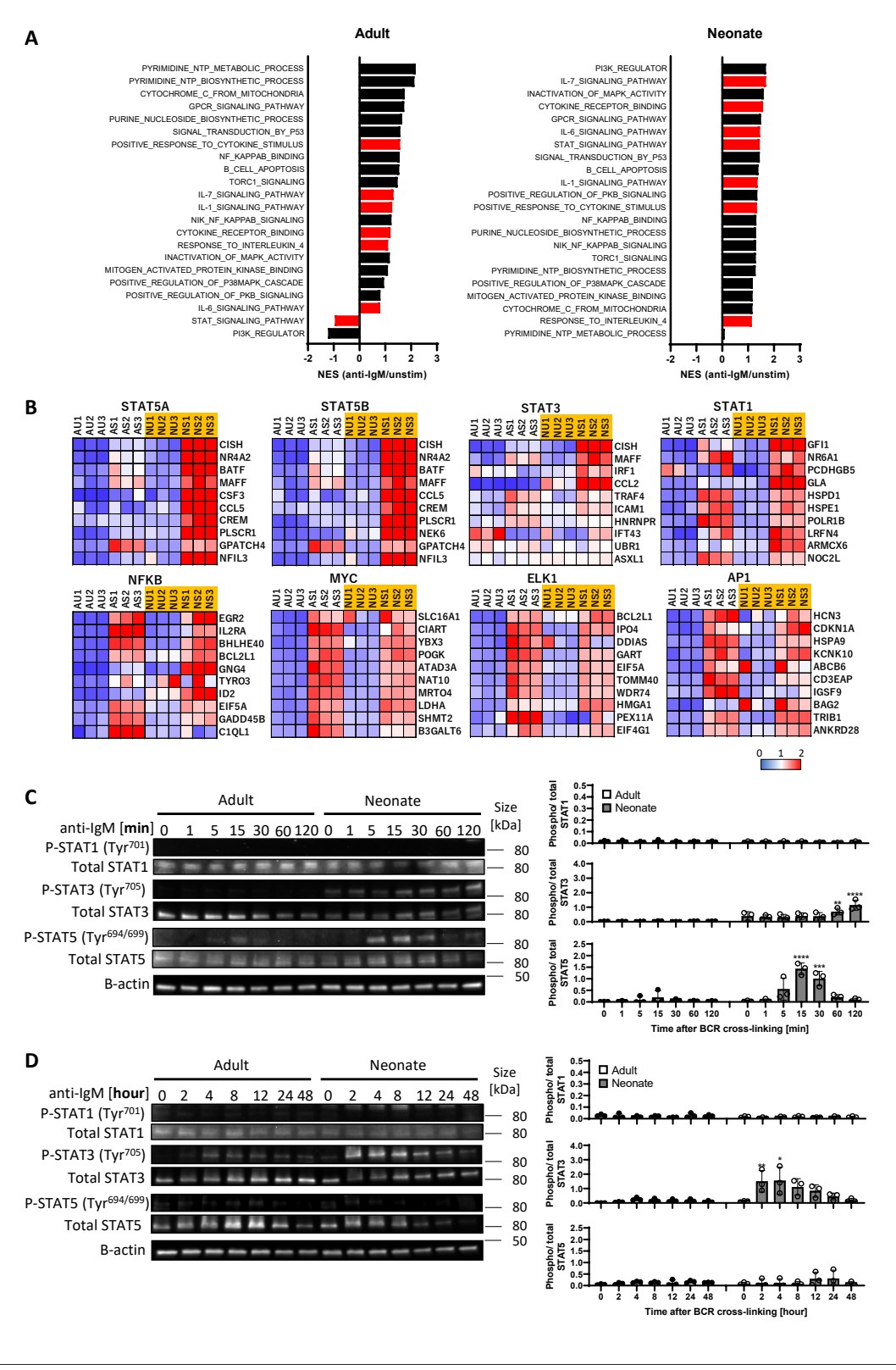

**Figure 2.** STAT3 and STAT5 are uniquely activated in neonatal B cells following BCR cross-linking. In all experiments, splenic CD19+CD43- B cells were isolated and stimulated with 10 µg/mL F(ab')₂ fragments of anti-IgM antibodies to engage BCR under different conditions. RNA-seq analysis was performed on CD19+CD43- cells stimulated with anti-IgM antibodies for 7 hr (biological triplicates). Total RNA was isolated for regular RNA

*Figure 2 continued on next page*

*Figure 2 continued*

sequencing. (**A**) Gene set enrichment analysis (GSEA) was performed using the hallmark gene sets in C5 gene ontology (GO). Normalized enrichment scores (NES) for representative gene sets from the C5 GO molecular signature databases that are correlated with the anti-IgM-stimulated phenotype (positive values) or unstimulated phenotype (negative values) following GSEA. Gene sets significantly enriched (FDR *q*-val <0.25) are ordered by the increasing NES. Bars in red indicate gene sets related to cytokine receptor and STAT signaling. (**B**) GSEA was performed using the hallmark gene sets in C3 transcription factor targets (TFT). Heat maps show top 10 genes in selected TFs. AU: adult unstimulated; AS: adult stimulated; NU: neonate unstimulated; NS: neonate stimulated. (**C and D**) CD19+CD43- cells were stimulated with anti-IgM antibodies for the indicated duration, and whole cell extracts were collected for immunoblot analysis of STAT1, STAT3, and STAT5. All experiments were repeated twice for reproducibility, and data are shown as the mean ± s.d. of three biological replicates. *P* values were calculated using one-way ANOVA with a Dunnett's multiple comparisons test (*p<0.05, **p<0.01, ***p<0.001, and ****p<0.0001).

The online version of this article includes the following source data and figure supplement(s) for figure 2:

**Source data 1.** Raw data of all western blots from *Figure 2*.

**Source data 2.** Complete and uncropped membranes of all western blots from *Figure 2*.

**Figure supplement 1.** STAT3 and STAT5 are uniquely activated in neonatal CD19+CD43 B cells following BCR cross-linking.

**Figure supplement 2.** STAT1 is phosphorylated in adult and neonatal splenic CD19+CD43 B cells in response to IL-21 but not to BCR cross-linking.

**Figure supplement 2—source data 1.** Raw data of all western blots from *Figure 2—figure supplement 2*.

**Figure supplement 2—source data 2.** Complete and uncropped membranes of all western blots from *Figure 2—figure supplement 2*.

**Figure supplement 3.** Akt, p38, JNK, and ERK phosphorylations in adult and neonatal splenic CD19+CD43 B cells following BCR cross-linking.

**Figure supplement 3—source data 1.** CRaw data of all western blots from *Figure 2—figure supplement 3*.

**Figure supplement 3—source data 2.** Complete and uncropped membrane of all western blots from *Figure 2—figure supplement 3*.

## STAT3 and STAT5 are involved in neonatal BCR-induced IL-10 production

We next asked whether the differential activation of STAT3 and STAT5 in BCR-stimulated adult and neonatal CD19+CD43- cells help explain elevated IL-10 production from neonatal CD19+CD43- cells. Although the involvement of STAT5 is not clear in IL-10 production, STAT3 has been shown to function as a promotor of IL-10 production in several types of cells, including human B cells (*Benkhart et al., 2000*; *Cheng et al., 2003*; *Stumhofer et al., 2007*; *Liu et al., 2014*). To assess the participation of these signaling molecules in IL-10 production, we used chemical inhibitors known to ablate STAT3 and STAT5 activation (*Nelson et al., 2011*; *Siddiquee et al., 2007*). We found that both S3I-201 (STAT3 inhibitor) and Pimozide (STAT5 inhibitor) inhibited BCR-induced *Il10* mRNA expression and IL-10 protein production (*Figure 3A, B*). To gain further insight into the activation of STAT3 and STAT5 downstream of BCR, we focused on Janus-activated kinases (JAKs) because STAT proteins are activated by JAKs in cytokine receptor signaling pathways (*Shuai and Liu, 2003*). We selected the pan-JAK inhibitor Pyridone 6 (*Thompson et al., 2002*) to test the involvement of STAT3 and STAT5. We first verified the inhibitory activity of Pyridone 6 in B cells, which effectively inhibited STAT3 and STAT5 phosphorylations induced by IL-6 and IL-21, respectively (*Figure 3—figure supplement 1A*). When neonatal B cells were stimulated through BCR in the presence of Pyridone 6, STAT3 phosphorylation, but not STAT5 phosphorylation, was completely suppressed (*Figure 3C*). Similarly, as previously observed (*Wang et al., 2007*), specific JAK2 inhibitor AG490 failed to inhibit STAT5 phosphorylation (*Figure 3—figure supplement 1B*). While neonatal BCR-induced STAT5 activation was JAK-independent, it was inhibited by Ibrutinib (Bruton's tyrosine kinase (BTK) inhibitor) and Staurosporine (protein kinase C (PKC) inhibitor) as shown previously (*Karras et al., 1996*; *Mahajan et al., 2001*; *Figure 3—figure supplement 1B*). These results suggested that whereas BCR induced STAT3 phosphorylation relied on JAKs, STAT5 is activated in a JAK-independent and BTK/PKC-dependent fashion.

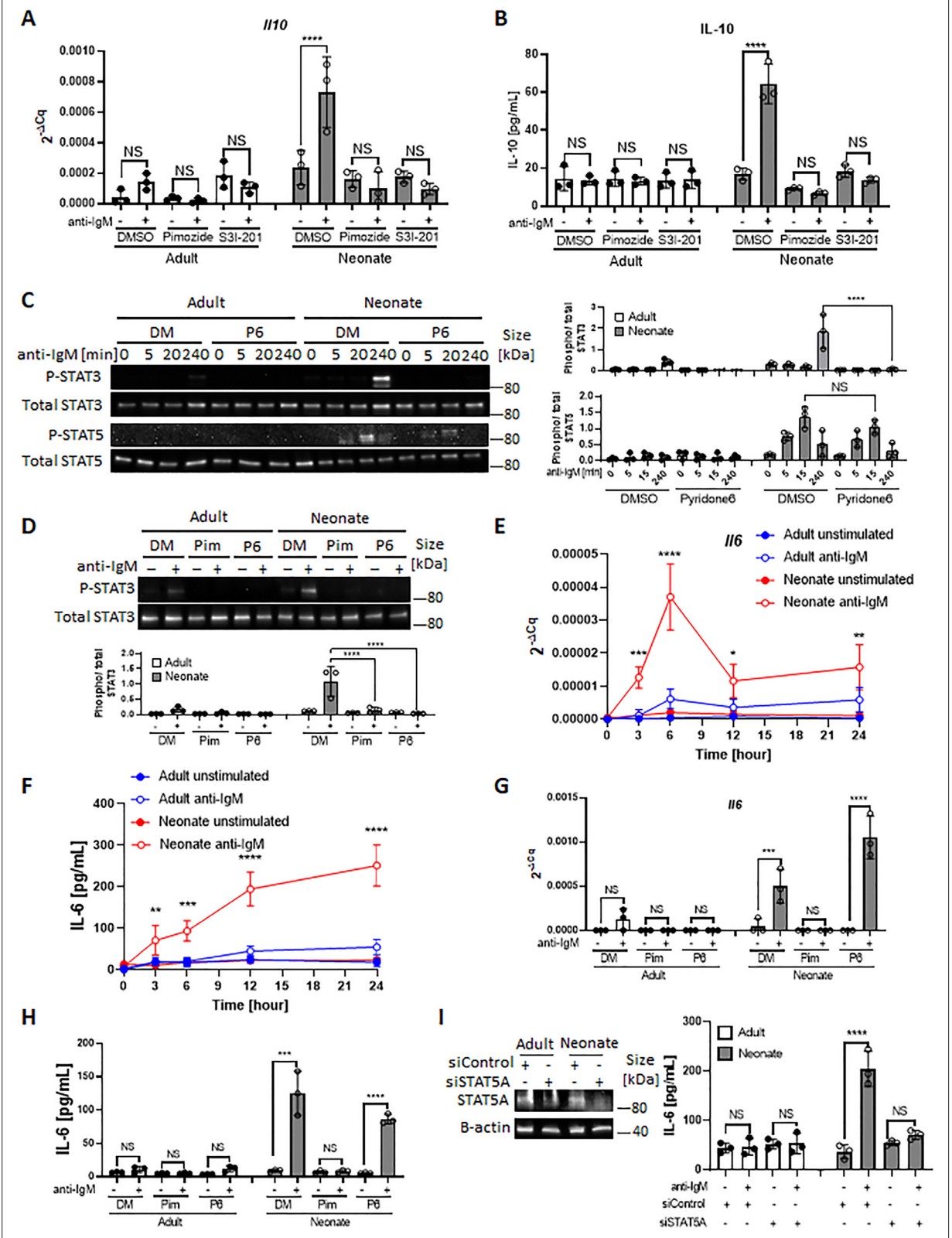

**Figure 3.** STAT3 is activated by autocrine IL-6 in a STAT5-dependent manner. In all experiments splenic CD19+CD43- B cells were isolated and stimulated with 10 μg/mL F(ab')₂ fragments of anti-IgM antibodies to engage BCR under different conditions. (**A and B**) CD19+CD43- cells were pre-treated with DMSO, 20 μM Pimozide, or 100 μM S3I-201 for 1 hr and then incubated in the absence or presence of anti-IgM antibodies for 7 hr (biological triplicate). *Il10* mRNA expression was determined by RT-qPCR (**A**) and the amount of secreted IL-10 was determined by ELISA (**B**). (**C**)

*Figure 3 continued on next page*

*Figure 3 continued*

CD19[+]CD43[-] cells were pre-treated with DMSO (DM) or 1 µM Pyridone 6 (**P6**) for 1 hr prior to stimulation with anti-IgM antibodies for the indicated duration and changes in STAT3 (Tyr[705]) and STAT5 (Tyr[694/699]) phosphorylations were detected in Western blot analysis (biological triplicate). (**D**) CD19[+]CD43[-] cells were pre-treated with DMSO, 20 µM Pimozide (Pim), or 1 µM Pyridone 6 (P6) for 1 hr and then incubated in the absence or presence of anti-IgM antibodies for 4 hr and changes in STAT3 (Tyr[705]) phosphorylation were detected in western blot analysis (biological triplicate). (**E and F**) CD19[+]CD43[-] cells were incubated with anti-IgM antibodies for the indicated duration and *Il6* mRNA expression was determined by RT-qPCR (**E**) and the amount of secreted IL-6 was measured by ELISA (**F**) (biological triplicate). *P* values (Adult anti-IgM vs Neonate anti-IgM) were calculated using two-way ANOVA. (**G and H**) CD19[+]CD43[-] cells were pre-treated with DMSO, 20 µM Pimozide, or 1 µM Pyridone 6 for 1 hr and then incubated in the absence or presence of anti-IgM antibodies for 4 hr and *Il6* mRNA expression was determined by RT-qPCR (**G**) and the amount of secreted IL-6 was determined by ELISA (**H**) (biological triplicate). (**I**) CD19[+]CD43[-] cells were transfected with siRNA targeting *STAT5A* (siSTAT5A) or non-targeting control siRNA (siControl) for 48 hr, and then incubated in the absence or presence of anti-IgM antibodies for 4 hr. The amount of STAT5A was examined by immunoblot analysis. The amount of secreted IL-6 was determined by ELISA (biological triplicate). Unless otherwise is indicated, *P* values were calculated using one-way ANOVA with a Dunnett's multiple comparisons test (*p<0.05, **p<0.01, ***p<0.001, and ****p<0.0001). No significant difference; NS. In all experiments data are shown as the mean ±s.d. of three biological replicates.

The online version of this article includes the following source data and figure supplement(s) for figure 3:

**Source data 1.** Raw data of all western blots from *Figure 3*.

**Source data 2.** Complete and uncropped membrane of all western blots from *Figure 3*.

**Figure supplement 1.** The effect of inhibitors on STAT3 and STAT5 phosphorylations in splenic CD19[+]CD43 cells.

**Figure supplement 1—source data 1.** Raw data of all western blots from *Figure 3—figure supplement 1*.

**Figure supplement 1—source data 2.** Complete and uncropped membrane of all western blots from *Figure 3—figure supplement 1*.

## Neonatal BCRs indirectly activate STAT3 via STAT5-dependent autocrine IL-6

Differences in the phosphorylation kinetics (*Figure 2*, B and C) and the dependency on JAKs (*Figure 3C*) between STAT3 and STAT5 in CD19[+]CD43[-] cells led us to hypothesize that STAT5 had a role in mediating neonatal BCR signaling to STAT3 in a JAK-dependent manner. To test this hypothesis, we investigated whether the STAT5 inhibitor Pimozide would block STAT3 phosphorylation. Supporting this hypothesis and suggesting that STAT5 functions as an upstream mediator of STAT3, Pimozide suppressed neonatal BCR-induced STAT3 phosphorylation (*Figure 3D*). Next, we sought to elucidate how STAT5 activates STAT3 in a JAK-dependent fashion. As JAKs are activated by cytokine receptors and several studies proposed autocrine IL-6 signaling in tumor cell lines (*Kawashima et al., 2001*; *Yeh et al., 2006*; *Gao et al., 2007*; *Hartman et al., 2011*; *Rodriguez-Barrueco et al., 2015*), we envisaged an autocrine activity of IL-6 produced by BCR-stimulated neonatal CD19[+]CD43[-] B cells. Indeed, we found significantly higher *Il6* mRNA expression and IL-6 protein production following BCR cross-linking in neonatal CD19[+]CD43[-] B cells compared to those in adult counterparts (*Figure 3E, F*). Higher IL-6 production in neonatal B cells following BCR cross-linking was also observed by flow cytometry (*Figure 3—figure supplement 1C*). While BCR-mediated signaling alone is not sufficient to induce IL-6 in adult B cells (*Kenny et al., 2013*; *Arkatkar et al., 2017*), studies in macrophages and cell lines have shown that STAT5 can contribute to IL-6 promoter activity (*Kawashima et al., 2001*; *Kimura et al., 2005*). To test whether the IL-6 production by neonatal B cells was also governed by STAT5 activation, we stimulated cells in the presence of the STAT5 inhibitor, Pimozide. Indeed, Pimozide abrogated both *Il6* mRNA expression and IL-6 protein production following BCR engagement, whereas the pan-JAK inhibitor Pyridone 6 did not (*Figure 3G, H*). This result was reproduced by using small interfering RNA targeting STAT5, which also suppressed BCR-induced IL-6 production (*Figure 3I*). These findings indicated that STAT5 has an essential role in the production of IL-6 by BCR-activated neonatal CD19[+]CD43[-] cells.

Next, we sought to investigate whether the STAT3 phosphorylation in BCR-activated neonatal cells was due to the IL-6 secreted from CD19[+]CD43[-] cells. IL-6 signaling is mediated by membrane-bound or soluble IL-6 receptor alpha (IL-6Rα) and a transmembrane protein gp130 (*Su et al., 2017*). Only after forming a complex with IL-6, the IL-6Rα associates with gp130 and transduces signals for JAK-STAT3 pathway (*Taga et al., 1989*; *Murakami et al., 1993*). Suggesting a role for IL-6 in STAT3 activation, blockade of IL-6 signaling with anti-IL-6Rα monoclonal antibody (15A7) reduced neonatal BCR-induced STAT3 phosphorylation, but not STAT5 phosphorylation (*Figure 4A*). Similarly, the gp130 inhibitor SC144 abrogated STAT3 phosphorylation following BCR engagement (*Figure 4—figure*

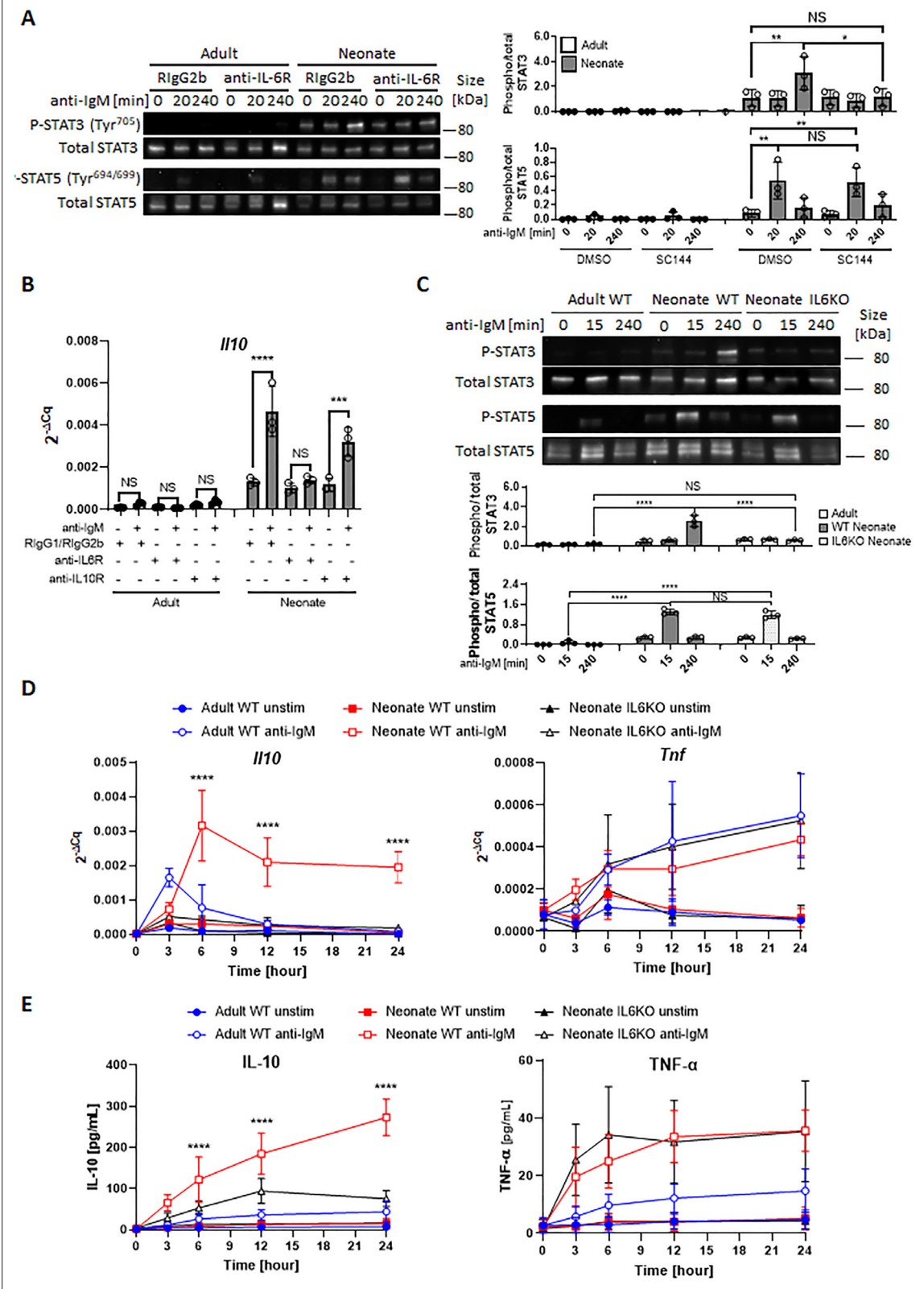

**Figure 4.** Autocrine IL-6 activates IL-10 expression in a STAT3-dependent manner. In all experiments, splenic CD19⁺CD43⁻ B cells were isolated and stimulated with 10 µg/mL F(ab')₂ fragments of anti-IgM antibodies to engage BCR under different conditions. (**A**) CD19⁺CD43⁻ cells were stimulated with anti-IgM antibodies for the indicated duration and changes in STAT3 (Tyr⁷⁰⁵) and STAT5 (Tyr⁶⁹⁴/⁶⁹⁹) phosphorylations were detected in Western blot analysis (biological triplicate). (**B**) CD19⁺CD43⁻ cells were pre-treated with 10 µg/mL isotype control antibodies (Rat IgG1/Rat IgG2b), anti-IL-6R antibody

*Figure 4 continued on next page*

*Figure 4 continued*

or anti-IL-10R antibody for 1 hr prior to stimulation with of anti-IgM antibodies for 7 h and *Il10* mRNA expression was determined by RT-qPCR (biological triplicate). (**C**) CD19⁺CD43⁻ cells, isolated from wild-type (WT) adult, WT neonate, and IL-6-deficient (IL6KO) neonate, were stimulated with anti-IgM antibodies for the indicated duration and changes in STAT3 (Tyr$^{705}$) and STAT5 (Tyr$^{694/699}$) phosphorylations were detected in Western blot analysis (biological triplicate). (**D and E**) CD19⁺CD43⁻ cells were incubated with anti-IgM antibodies for the indicated duration. *Il10* and *Tnf* mRNA expression were determined by RT-qPCR (**D**) and the amounts of secreted IL-10 and TNF-α were determined by ELISA (**E**) (biological triplicate). *P* values (Neonate WT anti-IgM vs Neonate IL6KO anti-IgM) were calculated using two-way ANOVA (****$p<0.0001$). Unless otherwise indicated, in other experiments *P* values were calculated using one-way ANOVA with a Dunnett's multiple comparisons test (*$p<0.05$, **$p<0.01$, ***$p<0.001$, and ****$p<0.0001$). No significant difference; NS. In all experiments, data are shown as the mean ±s.d. of three biological replicates.

The online version of this article includes the following source data and figure supplement(s) for figure 4:

**Source data 1.** Raw data of all western blots from *Figure 4*.

**Source data 2.** Complete and uncropped membrane of all western blots from *Figure 4*.

**Figure supplement 1.** Inhibition of gp130 blocks STAT3 activation in BCR stimulated neonatal cells.

**Figure supplement 1—source data 1.** Raw data of all western blots from *Figure 4—figure supplement 1*.

**Figure supplement 1—source data 2.** Complete and uncropped membrane of all western blots from *Figure 4—figure supplement 1*.

**Figure supplement 2.** Autocrine IL-10 has a minor role in STAT3 activation following BCR cross-linking.

**Figure supplement 2—source data 1.** Raw data of all western blots from *Figure 4—figure supplement 2*.

**Figure supplement 2—source data 2.** Complete and uncropped membrane of all western blots from *Figure 4—figure supplement 2*.

**Figure supplement 3.** Autocrine/paracrine IL-1β is not involved in neonatal BCR-induced IL-10 production.

**Figure supplement 3—source data 1.** Raw data of all western blots from *Figure 4—figure supplement 3*.

**Figure supplement 3—source data 2.** Complete and uncropped membrane of all western blots from *Figure 4—figure supplement 3*.

**Figure supplement 4.** Autocrine/paracrine IL-35 is not involved in neonatal BCR-induced IL-10 production.

*supplement 1*). While the blocking experiments indicated that IL-6 produced in response to BCR-engagement was responsible for STAT3 activation, STAT3 is also activated by IL-10 signaling (*Shouval et al., 2014*; *Figure 4—figure supplement 2A*). Since BCR stimulation also induces IL-10 production in neonatal B cells, we tested the involvement of IL-10 in STAT3 phosphorylation by using anti-IL-10R blocking monoclonal antibody. We found that anti-IL-10R antibody did not reduce neonatal BCR-induced STAT3 phosphorylation while anti-IL-6Rα antibody did (*Figure 4—figure supplement 2B*). These results suggest that neonatal CD19⁺CD43⁻ cells produced IL-6 following BCR cross-linking in a STAT5-dependent manner, and subsequently the secreted IL-6 activated STAT3 via the IL-6R-gp130 pathway in an autocrine/paracrine fashion.

## Neonatal B cells produce IL-10 in an autocrine fashion by IL-6-activated STAT3

The fact that both STAT3 and STAT5 inhibitors blocked *Il10* mRNA expression and IL-10 protein production from neonatal CD19⁺CD43⁻ cells (*Figure 3*, A and B) and the IL-6 produced by BCR-stimulated CD19⁺CD43⁻ cells was responsible for the activation of STAT3 (*Figure 4A*) suggested that IL-10 might be produced in response to IL-6. To begin addressing this question, we first investigated whether the production of IL-10 was dependent on IL-6 signaling in CD19⁺CD43⁻ cells. We found that blocking with antibodies against IL-6Rα, but not with anti-IL-10R, totally abrogated *Il10* mRNA expression 7 hr after BCR stimulation (*Figure 4B*). To further elucidate this question, we evaluated responses in splenic CD19⁺CD43⁻ B cells from IL-6-deficient neonatal mice. We began by assessing the activation of STATs in IL-6 knock out (KO) neonatal splenic CD19⁺CD43⁻ cells. Further underscoring the dependency of BCR-induced STAT3 activation on IL-6 production, BCR cross-linking did not induce STAT3 phosphorylation in CD19⁺CD43⁻ cells from IL-6 KO neonatal mice, whereas STAT5 phosphorylation was comparable between wild-type and IL-6 KO strains (*Figure 4C*). We excluded the possibility of a defect in STAT3 activation in IL-6-deficient CD19⁺CD43⁻ cells because recombinant IL-6 induced the phosphorylation of STAT3 in IL-6 KO neonatal CD19⁺CD43⁻ cells (*Figure 4—figure supplement 2C*).

Next, we assessed the cytokine levels in IL-6-deficient neonatal B cells following BCR engagement. Both *Il10* gene expression and IL-10 protein levels were significantly reduced in IL-6 KO neonatal CD19⁺CD43⁻ cells compared to wild-type counterparts (*Figure 4*, D and E). IL-6 deficiency did not create an overall suppressive state because both *Tnf* gene expression and TNF-α protein levels

were comparable between wild-type and IL-6-deficient neonatal CD19+CD43- cells (*Figure 4*, D and E). Notably, despite the significant reduction in IL-10 levels, there were residual gene and protein expression in CD19+CD43- cells from IL-6 KO mice. To test whether other cytokines also contribute to IL-10 production, we focused on IL-1β because IL-1β is shown to enhance IL-10 production by CD40-activated B cells (*Rosser et al., 2014*). While neonatal CD19+CD43- cells significantly increased *Il1b* gene expression following BCR cross-linking (*Figure 4—figure supplement 3A*), blocking IL-1β signaling using a neutralizing antibody failed to reduce IL-10 production by IL-6-deficient neonatal CD19+CD43- B cells (*Figure 4—figure supplement 3B, C*). In addition, biologically active recombinant IL-1β (*Figure 4—figure supplement 3D*) did not induce IL-10 while recombinant IL-6 did (*Figure 4—figure supplement 3E*). The IL-12 family member cytokine IL-35 has been shown to induce Breg differentiation and IL-10 production (*Wang et al., 2014*). The autocrine effect of IL-35 on IL-10 production by Bregs has been considered since this cytokine is secreted by Bregs as well as Tregs (*Huang et al., 2017*). To assess whether IL-35 may be involved in IL-10 secretion, we measured IL-35 production from anti-IgM stimulated neonatal B cells. Ruling out its involvement, IL-35 production was not observed following BCR cross-linking (*Figure 4—figure supplement 4*). These results suggested that molecule(s) other than IL-1β and IL-35 are likely responsible for the production of residual IL-10 from BCR-stimulated IL-6 KO neonatal CD19+CD43- cells.

In addition to the elevated IL-6 production, flow cytometry assessment revealed that neonatal B cells expressed modest but significantly higher levels of surface IL-6Rα compared to their adult counterpart (*Figure 5A*). Measurement of *il6ra* expression in B cells from both age groups also showed that neonatal cells expressed higher *il6ra* but this difference did not reach statistical significance (*Figure 5—figure supplement 1*). The expression level of gp130 was comparable between the two age groups (*Figure 5—figure supplement 2*). To test whether higher IL-6Rα expression translated into enhanced IL-6 activity, we incubated purified CD19+CD43- population in the presence of increasing concentrations of recombinant IL-6 and measured P-STAT3 in western blot analysis. We detected P-STAT3 in neonatal cells with as little as 0.04 ng/mL of IL-6 whereas approximately 62 times more IL-6 (2.5 ng/mL) was needed to induce the phosphorylation of STAT3 in adult cells (*Figure 5B*). Quantification of the IL-6 activity by computing half maximal effective concentration ($EC_{50}$) indicated that the $EC_{50}$ for neonatal B cells (0.98±0.41 ng/mL) was significantly lower than that for adult counterparts (4.37±1.27 ng/mL). Moreover, the elevated neonatal B cell P-STAT3 activity translated into increased IL-10 production in IL-6-stimulated neonatal CD19+CD43- cells (*Figure 5C*).

Finally, we wanted to examine whether IL-6 was also involved in the production of IL-10 in response to toll-like receptor (TLR) ligands because neonatal B cells have also been shown to produce elevated levels of IL-10 in response to TLR ligand stimulation (*Yanaba et al., 2008*; *Walker and Goldstein, 2007*; *Andrade et al., 2013*). As reported previously, TLR9 and TLR4 ligands induced high levels of IL-10 from wild-type neonatal mouse CD19+CD43- cells (*Figure 5—figure supplement 3*). IL-10 secreted by IL-6-deficient neonatal CD19+CD43- cells were comparable to that from wild-type neonatal CD19+CD43- cells in response to CpG and lipopolysaccharide (LPS), suggesting that TLR signaling-induced IL-10 production was not dependent on IL-6. Taken together, these results revealed a mechanism whereby neonatal CD19+CD43- cell BCR-STAT5 axis-induces IL-6 which in turn activates STAT3 in autocrine/paracrine fashion, leading to IL-10 production (*Figure 5D*).

## Neonatal BCR activates STAT5 via protein kinase C, focal adhesion kinase, and Rac1

We next sought to elucidate how neonatal BCRs activate STAT5 in a JAK-independent manner (*Figure 3C*). We focused on the involvement of protein kinase C (PKC), focal adhesion kinase (FAK) and Rac1 because PKC was suggested to play a role in BCR-triggered STAT5 activation (*Karras et al., 1996*) and autophosphorylation of FAK and Rac1 mediates STAT5 activation in oncogenic cells (*Chatterjee et al., 2014*). Moreover, PKC has been shown to promote FAK phosphorylation (*Vuori and Ruoslahti, 1993*; *Lewis et al., 1996*; *Heidkamp et al., 2003*) and to activate Rac1 (*Nomura et al., 2007*; *Tu et al., 2020*). We found that BCR cross-linking increased the phosphorylation of PKC in neonatal CD19+CD43- cells, but not in adult cells (*Figure 6A*). We assessed the two phosphorylation sites of FAK: an autophosphorylation site (Tyr[397]) that functions as a Src homology 2 (SH2) binding site and the kinase domain (Tyr[567]) that promotes FAK catalytic activity (*Lawson and Schlaepfer, 2013*). FAK was constitutively phosphorylated at Tyr[397] in neonatal cells and this phosphorylation increased

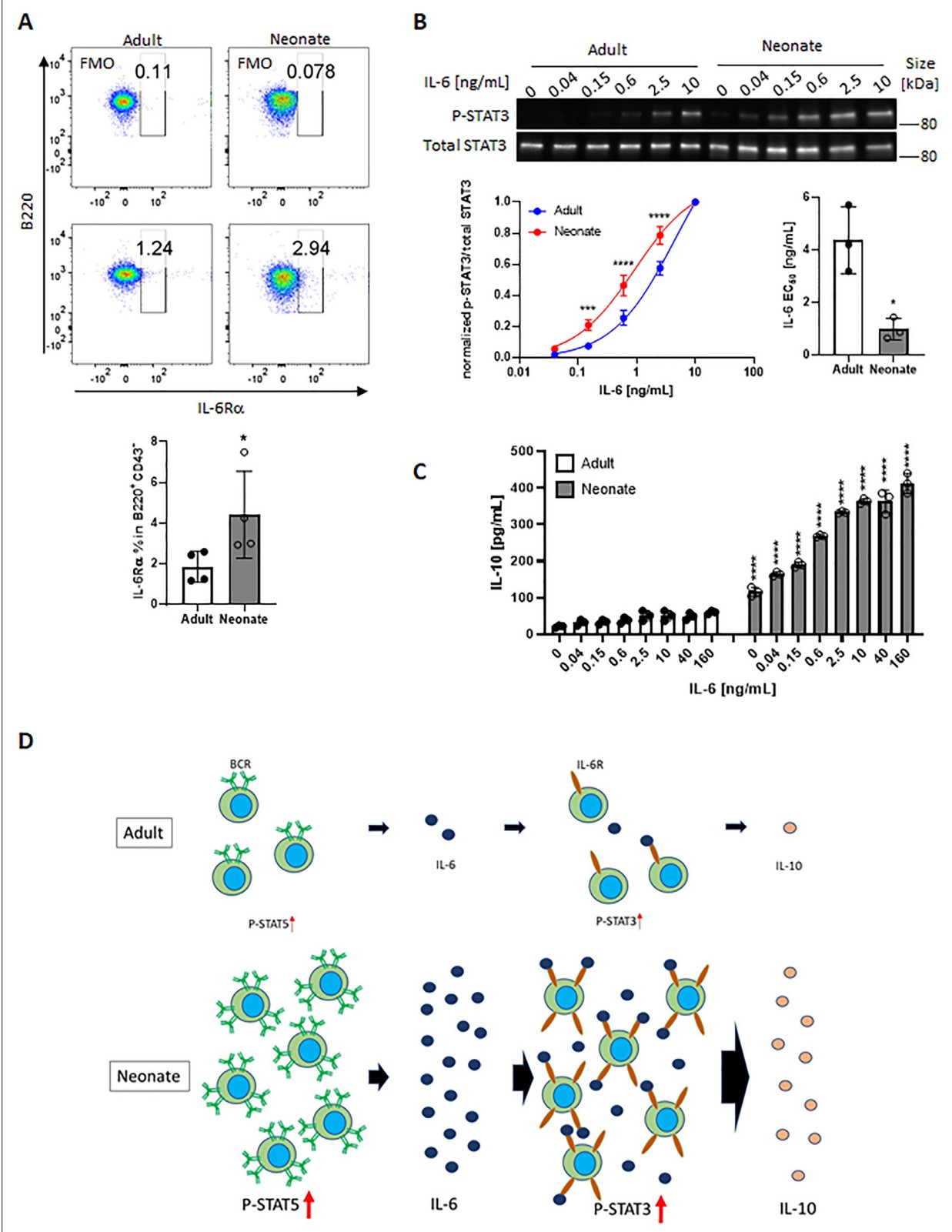

**Figure 5.** Neonatal BCRs activate STAT5 in PKC/FAK/Rac1/Syk and Btk signaling axis. In all experiments, isolated splenic CD19+CD43- B cells were used. (**A**) CD19+CD43- cells were analyzed for surface levels of IL-6Rα. Data are shown as the mean ± s.d. of three biological replicates. *P* values versus adult counterparts were calculated using two-tailed unpaired *t*-tests (*p<0.05). (**B**) CD19+CD43- cells were stimulated with recombinant IL-6 at indicated concentrations for 30 min. Changes in STAT3 (Tyr705) phosphorylation were detected in western blot analysis. Data are representative

*Figure 5 continued on next page*

*Figure 5 continued*

of three biological replicates. Data are shown as the mean ±s.d. of normalized ratio of P-STAT3 to total STAT3 in three biological replicates. *P* values versus adult counterparts were calculated using two-way ANOVA with a Sidak's multiple comparisons test for the dose-response curve (***p<0.001 and ****p<0.0001) and two-tailed Student's *t*-test for the IL-6 EC$_{50}$ comparison (*p<0.05). (**C**) CD19+CD43- cells were stimulated with increasing concentrations of recombinant IL-6 for 48 hr. The amount of secreted IL-10 was determined by ELISA. Data are shown as the mean ±s.d. of two biological replicates. *P* values versus adult counterparts were calculated using one-way ANOVA with a Dunnett's multiple comparisons test (****p<0.0001). (**D**) Schematic representation of cellular events leading to enhanced IL-10 production from neonatal CD19+CD43- B cells following BCR cross-linking. Larger population of neonatal CD19+CD43- B cells activate STAT5 through highly expressed IgM, and STAT5 causes IL-6. Large amount of IL-6 and highly expressed IL-6Rα on neonatal CD19+CD43- B cells synergistically activate STAT3 in an autocrine/paracrine fashion, leading to IL-10 production.

The online version of this article includes the following source data and figure supplement(s) for figure 5:

**Source data 1.** Raw data of all western blots from *Figure 5*.

**Source data 2.** Complete and uncropped membrane of all western blots from *Figure 5*.

**Figure supplement 1.** Comparable expression of gp130 on adult and neonatal B cells.

**Figure supplement 2.** Comparable expression of gp130 on adult and neonatal B cells.

**Figure supplement 3.** TLR-induced IL-10 production is not dependent on IL-6.

further after stimulation, whereas BCR cross-linking triggered a comparable increase in phosphorylation at Tyr[567] in adults and neonates (*Figure 6A* and *Figure 6—figure supplement 1*), suggesting that the FAK catalytic activity did not lead to the unique STAT5 phosphorylation in neonates. Confirming their upstream role in controlling STAT5 activation, both the PKC inhibitor staurosporine and the FAK inhibitor 14 (F-14) inhibited neonatal BCR-induced STAT5 phosphorylation (*Figure 6B*). Moreover, *Il6* gene expression in response to neonatal BCR engagement was impeded by these inhibitors as well as the STAT5 inhibitor (*Figure 6C*). Also, the Rac1 inhibitor, NSC23766 inhibited STAT5 phosphorylation (*Figure 6D*) and *Il6* gene expression (*Figure 6E*) following BCR cross-linking. Having established that PKC governs BCR-induced STAT5 phosphorylation, we tested whether the specific PKC activator phorbol 12-myristate 13-acetate (PMA) also triggered STAT5 phosphorylation. As expected, PKC was phosphorylated in both adult and neonatal CD19+CD43- cells after PMA stimulation (*Figure 6—figure supplement 2A*). Also, PMA induced the phosphorylation of p65 and IκBα in both neonatal and adult CD19+CD43- cells (*Figure 6—figure supplement 2A*). However, only neonatal B cells manifested STAT5 phosphorylation in response to PMA (*Figure 6—figure supplement 2A*). Finally, by using specific inhibitors, we found that spleen tyrosine kinase (Syk) and Btk control BCR-induced STAT5 phosphorylation (*Figure 6—figure supplement 2B*). Together, these results unveiled the upstream pathways involving Syk, Btk, PKC, FAK, and Rac1 in BCR-induced *Il6* gene expression via STAT5 phosphorylation.

## IL-6 is not responsible for IL-10 production by CD43-expressing B-1 subset in neonatal mouse

Using a range of experiments, we demonstrated that BCR-crosslinking of CD19+CD43- B cells lead to IL-6 production and this IL-6 is responsible for IL-10 secretion from CD19+CD43- B cells (*Figure 4B, D and E*). Among the neonatal splenic B cells, the highest IL-10-producing subset is the CD43-expressing B-1 subset (*Figure 1B*). To determine whether IL-6 is also responsible for IL-10 production by this subset, we compared IL-10-producing B-1 subset from BCR-stimulated wild-type and IL-6 KO neonatal B cells. We found that the BCR cross-linking did not increase CD19+CD43+IL-10+ population in both wild-type and IL-6 KO cells (*Figure 6—figure supplement 3*). Thus, BCR-induced IL-6 does not promote the expression of IL-10 by B-1 population.

## IL-10 secreted from neonatal CD19+CD43- cells suppress TNF-α production by macrophages

Bregs exert their immune suppressive functions via both IL-10-dependent and -independent mechanisms (*Floudas et al., 2016*). IL-10 derived from Bregs has a major role in inhibiting inflammatory cytokine production by monocytes (*Iwata et al., 2011*; *Ray et al., 2012*). Neonatal B cells have been shown to restrict Th1 responses in vivo and ex vivo by suppressing myeloid cell functions in an IL-10-dependent manner (*Zhang et al., 2007*; *Sun et al., 2005*; *Walker and Goldstein, 2007*). We next

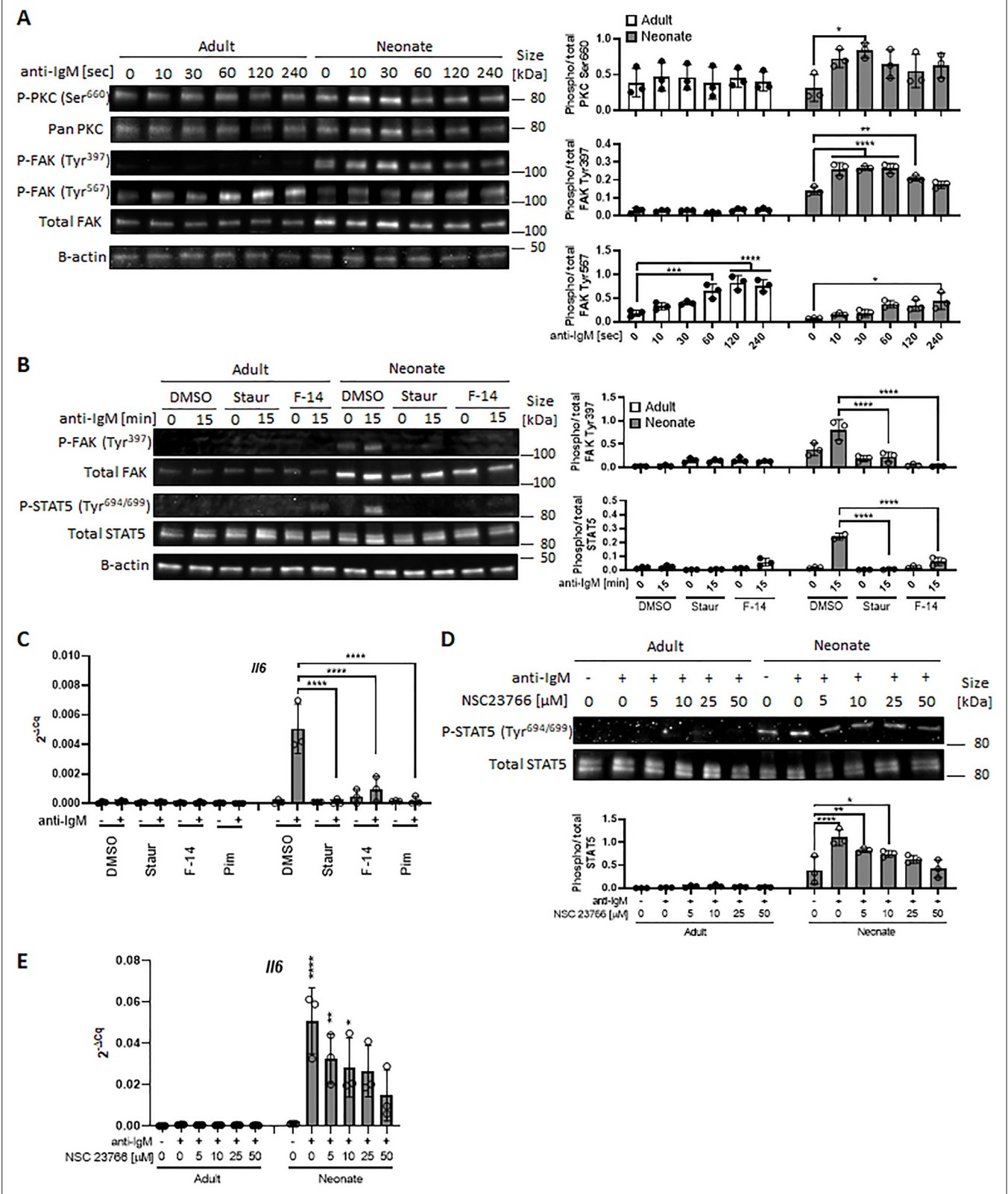

**Figure 6.** In all experiments, splenic CD19⁺CD43⁻ B cells were isolated and stimulated with 10 µg/mL F(ab')₂ fragments of anti-IgM antibodies to engage BCR under different conditions. (**A**) CD19⁺CD43⁻ cells were stimulated with anti-IgM antibodies for the indicated duration and changes in PKC (Ser660), FAK (Tyr397), and FAK (Tyr567) phosphorylations were detected in western blot analysis (biological triplicate). (**B**) CD19⁺CD43⁻ cells were pre-treated with DMSO, 20 nM PKC inhibitor Staurosporine (Staur), or 10 µM FAK inhibitor 14 (F-14) for 1 hr and then stimulated with anti-IgM antibodies

*Figure 6 continued on next page*

*Figure 6 continued*

for the indicated duration. Changes in FAK (Tyr[397]) and STAT5 (Tyr[694/699]) phosphorylations were detected in Western blot analysis (biological triplicate). (**C**) Isolated CD19[+]CD43[-] B cells were pre-treated with DMSO, 20 nM Staurosporine, or 10 µM F-14, or 20 µM Pimozide for 1 hr prior to incubation in the absence or presence of anti-IgM antibodies for 4 hr and *Il6* mRNA expression was determined by RT-qPCR (biological triplicate). (**D**) CD19[+]CD43[-] cells were pre-treated with Rac1 inhibitor NSC23766 for 1 hr and then stimulated with anti-IgM antibodies for 15 min. Changes in STAT5 (Tyr[694/699]) phosphorylation was detected in western blot analysis (biological triplicate). (**E**) Isolated CD19[+]CD43[-] B cells were pre-treated with NSC23766 for 1 hr prior to incubation in the absence or presence of anti-IgM antibodies for 20 hr and *Il6* mRNA expression was determined by RT-qPCR (biological triplicate). In all experiments, data shown as the mean ± s.d. of three biological replicates. *P* values were calculated using one-way ANOVA with a Dunnett's multiple comparisons test (*p<0.05, **p<0.01, ***p<0.001, and ****p<0.0001).

The online version of this article includes the following source data and figure supplement(s) for figure 6:

**Source data 1.** Raw data of all western blots from *Figure 6*.

**Source data 2.** Complete and uncropped membrane of all western blots from *Figure 6*.

**Figure supplement 1.** Neonatal BCRs activate FAK.

**Figure supplement 2.** BCR-induced STAT5 activation is dependent on Syk, Btk, and PKC.

**Figure supplement 2—source data 1.** Raw data of all western blots from *Figure 6—figure supplement 2*.

**Figure supplement 2—source data 2.** Complete and uncropped membrane of all western blots from *Figure 6—figure supplement 2*.

**Figure supplement 3.** IL-6 is not responsible for IL-10 production by CD43-expressing B1 subset in neonatal mouse.

sought to determine whether neonatal BCR-induced autocrine IL-6 signaling has a role in the IL-10-dependent regulatory function of neonatal CD19[+]CD43[-] cells. To this end, conditioned media (CM) were prepared from BCR-stimulated splenic CD19[+]CD43[-] cells isolated from adult, wild-type neonatal (WT), and IL-6 KO neonatal mice and the inhibitory potential of the CM from each condition was tested by measuring TNF-α production from adult mouse peritoneal macrophages (*Figure 7—figure supplement 1*). We confirmed that CM from WT neonatal CD19[+]CD43[-] cells contained the highest amount of IL-10 (*Figure 7A*), whereas all three CMs contained low amount of TNF-α (*Figure 7B*). When incubated with macrophages, unstimulated cell-CMs from all three groups failed to induce

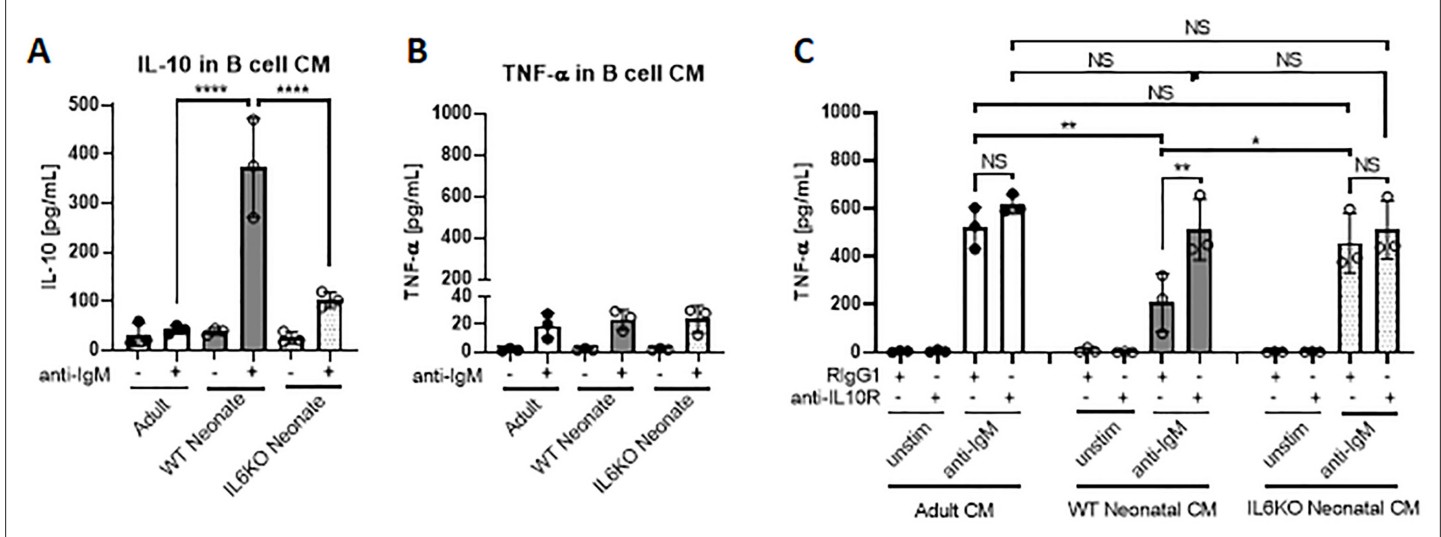

**Figure 7.** IL-10 secreted from neonatal CD19[+]CD43 B cells suppress TNF-α secretion from macrophages. (**A and B**) Isolated splenic CD19[+]CD43[-] B cells were stimulated with 10 µg/mL F(ab')₂ fragments of anti-IgM antibodies for 24 hr. IL-10 (**A**) and TNF-α (**B**) levels in conditioned medium (CM) were determined by ELISA. (**C**) Peritoneal macrophages isolated from adult mice were treated with 10 µg/mL anti-IL10R or isotype (Rat IgG1) control antibodies for 1 hr and then cultured in CM for 24 hr. Levels of TNF-α secreted by macrophages cultured in CM were determined by ELISA. Data are shown as the mean ± s.d. of three biological replicates. *P* values were calculated using one-way ANOVA with a Dunnett's multiple comparisons test (*p<0.05, **p<0.01, and ****p<0.0001).

The online version of this article includes the following figure supplement(s) for figure 7:

**Figure supplement 1.** Schematic description of experimental design to test the suppressive effect of IL-10 secreted from neonatal CD19[+]CD43 B cells.

**Figure supplement 2.** Schematic representation of IL-10 production in BCR stimulated neonatal CD19[+]CD43 cells.

TNF-α production (*Figure 7C*). In the presence of control antibody (Rat IgG1), CMs from anti-IgM-stimulated adult and IL-6 KO neonatal cells elicited significant and comparable increases in TNFα production (*Figure 7C*). In sharp contrast, CM from anti-IgM-stimulated wild-type neonatal cells triggered a blunted TNF-α production from macrophages. To test whether the low TNF-α secretion with CM from wild-type neonate was due to elevated IL-10 in the CM, we included blocking antibodies against IL-10R in the culture system. Suggesting a suppressive role for the elevated IL-10 levels in the CM from wild-type neonatal CD19+CD43- cells, IL-10R antibodies significantly increased the TNF-α production and restored its concentration to macrophages incubated with neonatal IL-6 KO CM. Since adult and IL-6 KO CM contained low IL-10 levels, addition of anti-IL-10R antibodies did not significantly alter the TNF-α production in macrophages incubated with these CMs. Thus, these results demonstrate a major role for autocrine IL-6 signaling in the suppressive effect of BCR-induced IL-10 from neonatal CD19+CD43- B cells on macrophage inflammatory cytokine production.

## Discussion

The propensity of neonatal splenic B cells to produce IL-10 with suppressive properties is well established (*Burdin et al., 1997*). The increased IL-10 production by neonatal splenic B cells is primarily attributed to the B-1 cells, which constitute approximately 30% of B cells in neonatal spleen compared to less than 5% in adult spleen (*Hayakawa et al., 1983*). Here, we described an additional subset of splenic B cells in neonates with higher IL-10 production potential than their adult counterparts. We found that BCR stimulation triggered IL-10 production by neonatal CD19+CD43- B cells. In CD19+CD43- cells, BCR stimulation led to STAT5 activation under the control of Syk, Btk, PKC, FAK, and Rac1. Interestingly, BCR-induced STAT5 activation did not directly lead to IL-10 expression. Instead, the increase in IL-10 production was mostly dependent on STAT5-mediated IL-6 secretion, which promoted IL-10 secretion in an autocrine and paracrine fashion.

The IL-10-dependent suppression of inflammation by B10 cells is well documented in adult mouse autoimmune, allergy and infection models (*Yanaba et al., 2008*; *Amu et al., 2010*; *Jeong et al., 2012*; *Horikawa et al., 2013*; *Jeong et al., 2016*). Despite the well-established requirement for antigen-mediated BCR signaling in B10 cell development, BCR-stimulation is not sufficient to induce the production of IL-10 from adult B10 cells (*Yanaba et al., 2009*). In fact, not only anti-IgM stimulation does not induce the secretion of IL-10 from adult B10 cells, but it also inhibits IL-10 production from LPS- and CD40L-stimulated B10 cells (*Yanaba et al., 2009*). Our results confirm the blunted IL-10 secretion from anti-IgM-stimulated adult B10 cells. Although neonatal B10 cells were reported previously by Yanaba and colleagues, the production of IL-10 from these cells in response to anti-IgM stimulation has not been studied (*Yanaba et al., 2009*). Our data show that neonatal CD19+CD43- B cells are highly sensitive to BCR stimulation, but the secretion of IL-10 depends on the production of IL-6 from these cells rather than directly linking BCR to the transcription of *Il10* (*Figure 7—figure supplement 2*). In contrast to CD19+CD43- B cells, B-1 cells secrete IL-10 spontaneously and the secretion of IL-10 is further increased following BCR-stimulation (*Alhakeem et al., 2015*). The BCR-mediated IL-10 production from B-1 cells depends on p38 MAPK activity (*Alhakeem et al., 2015*). The absence of IL-10 production directly after BCR stimulation in neonatal CD19+CD43- B cells in our experiments is not unlikely to be due to the absence of p38 MAPK activity in these cells, because adult CD19+CD43- B cells also failed to secrete IL-10 despite the phosphorylation of Akt, p38, JNK and ERK in response to BCR engagement. We found that the rapid IL-6 secretion following BCR engagement requires STAT5 activation (*Figure 7—figure supplement 2*). We also demonstrated that neonatal SH2 binding site (Tyr$^{397}$) on FAK was constitutively phosphorylated, and this phosphorylation was required for STAT5 phosphorylation because the FAK inhibitor, F-14 which inhibits phosphorylation at Tyr$^{397}$ effectively blocked BCR-induced STAT5 phosphorylation. These results indicated that FAK scaffold functions were essential to mediate BCR signaling to STAT5 in neonatal B cells as has been shown for FLT3 receptor signaling in oncogenic cells (*Chatterjee et al., 2014*). A member of Rho family of GTPases, Rac1 has been shown to have a role in STAT5 activation in several types of cells (*Benitah et al., 2003*; *Kawashima et al., 2006*) and Rac1 functions as a downstream effector of FAK (*Chatterjee et al., 2014*). Using specific inhibitors, we determined that Rac1, Syk, Btk, and PKC controlled neonatal BCR-induced STAT5 activation. Taken together, our data unveiled the signaling pathway mediated by Syk, Btk, PKC, FAK, and Rac1 in the activation of STAT5 and the production of IL-6 in BCR stimulated neonatal CD19+CD43- B cells (*Figure 7—figure supplement 2*).

Induction of IL-10 secretion from Breg cells by IL-6 has been shown previously by Rosser and colleagues in adult mice (*Rosser et al., 2014*). However, in this study the authors have discovered that IL-6, as well as IL-1β, produced by macrophages in response to microbiota mediated stimuli act on splenic and lymph node Breg cells to produce IL-10. An important distinction between our findings in neonatal CD19+CD43- B cells and those of Rosser and colleagues in adult Breg cells is that whereas microbiota-derived IL-6 and IL-1β required simultaneous CD40 signaling to induce IL-10 production from adult Breg cells, IL-6 alone was able to stimulate the production of IL-10 from neonatal CD19+CD43- B cells. The independence of neonatal CD19+CD43- B cells from CD40 signaling in IL-6-induced IL-10 secretion has in vivo relevance because neonatal B cells do not get activated through CD40 due to weak CD40L expression in neonatal T cells (*Durandy et al., 1995*; *Fuleihan et al., 1994*). As was observed by Rosser and colleagues (*Rosser et al., 2014*), we also found that recombinant IL-6 was not able to induce IL-10 production from adult CD19+CD43- B cells. We think that the heightened sensitivity of neonatal B cells to IL-6 is likely multifactorial. We found modestly elevated expression of IL-6R on neonatal CD19+CD43- B cells compared to adult cells. In addition to higher IL-6R expression, there may be additional enhancers downstream of neonatal IL-6R that may help in increased STAT3 phosphorylation and IL-10 production. Higher IL-6 production by neonatal B cells following BCR engagement may have implications other than suppression through increased IL-10 secretion. The beneficial effect of IL-6 in the generation of adult T follicular helper (Tfh) cells, which help activate germinal center B cells to differentiate into plasma cells and memory B cells in response to immunization and infection is well recognized (*Korn and Hiltensperger, 2021*; *Papillion et al., 2019*). We have previously shown that, in contrast to adults, IL-6 suppresses vaccine response in neonatal mice by inhibiting Tfh generation and expanding the suppressive T follicular regulatory helper (Tfr) cells (*Yang et al., 2018*). Thus, in addition to stimulating IL-10 production, increased IL-6 secretion from BCR-stimulated neonatal B cells may be blunting Tfh generation.

The IL-10 produced by neonatal BCR-stimulated CD19+CD43- B cells was functional because it inhibited TNF-α production by macrophages. We demonstrated that CM from anti-IgM-stimulated neonatal and adult B cells produced molecules that induced TNF-α production by macrophages, but this induction was blunted with the IL-10 in neonatal CM because the inhibitory anti-IL-10R antibody effectively restored the TNF-α secretion from macrophages. Moreover, there was no inhibition of TNF-α production from macrophages when CM from IL-6 KO B cells were used. In summary, our study revealed a detailed molecular and cellular mechanism involved in BCR-mediated IL-10 production by neonatal CD19+CD43- B cells (*Figure 7—figure supplement 2*). The lower threshold of producing IL-10 in response to BCR stimulation is likely amplified by the fact that neonatal CD19+CD43- B cells express higher levels of IgM than their adult counterparts. One possible biological reason for the elevated production of the suppressive cytokine IL-10 could be to dampen the inflammatory response when newborns encounter massive amounts of microbial stimuli after birth. This unique inhibitory pathway involving intermediate IL-6 secretion expands our understanding of the overall weak responses of neonates to vaccines and their susceptibility to infections.

## Materials and methods

### Mice

C57BL/6 J mice and IL-6 KO (B6.129S2-*Il6*$^{tm1Kopf}$/J) mice were purchased from The Jackson Laboratory and maintained in local facilities. Two- to 10-month-old male and female were used in mating. All mice were fed regular chow in a pathogen-free environment. The mouse experiments described in this study were performed in accordance with the US Food and Drug Administration/Center for Biologics Evaluation and Research Institutional Animal Care and Use Committee guidelines (permit 2002–31 and 2017–45).

### Cell isolation and culture

B lymphocytes were isolated from spleen tissues obtained from mixed sex neonates (6–8 days old) and adult (8–10 weeks old) female mice. Total splenic B cells (CD19+) were purified using CD19 MicroBeads (130-121-301, Miltenyi Biotec, San Jose, CA), and CD19+CD43- non-B-1 cells were purified using Mouse B Cell Isolation Kit (130-090-862, Miltenyi Biotec) which contains anti-CD43 antibody. The purity of isolated cells was assessed as B220+ and CD19+ double positive population with

flow cytometry. Peritoneal macrophages were isolated from adult (8–10 weeks old) female mice using Peritoneum Macrophage Isolation Kit (130-110-434, Miltenyi Biotec). Isolated cells were cultured in complete RPMI +GlutaMAX (Thermo Fisher Scientific, Waltham, MA) supplemented with 10% fetal bovine serum (Life Technologies, Frederick, MD), 2 mM glutamine (Life Technologies), 100 U/mL penicillin and 100 µg/ml streptomycin (Life Technologies), 1 mM sodium pyruvate (Life Technologies), 10 µM 2-mercaptoethanol (Sigma Aldrich, St Louis, MO), 20 mM HEPES (Life Technologies), and 1 mM MEM nonessential amino acids (Life Technologies). Cells were maintained in an incubator at 37 °C, 5% $CO_2$. Prior to stimulation, cells were treated with Pyridone 6 (Tocris Bioscience, Minneapolis, MN), Pimozide (Tocris Bioscience), S3I-201 (Sigma Aldrich), Staurosporine (Sigma Aldrich), FAK Inhibitor 14 (F-14) (R&D Systems, Minneapolis, MN), NSC23766 (R&D Systems), AG490 (Tocris Bioscience), SC144 (Tocris Bioscience), Syk Inhibitor (CAS622387-85-3, Sigma Aldrich), Ibrutinib (PCI-32765, Selleck Chemicals, Houston, TX), InVivoPlus rat IgG1 (HRPN, BioXcell, Lebanon, NH), InVivoMAb rat IgG2b (LTF-2, BioXcell), InVivo polyclonal Armenian hamster IgG (BioXcell), InVivoMAb anti-mouse IL-6R (15A7, BioXcell), InVivoPlus anti-mouse IL-10R (1B1.3A, BioXcell) or InVivoMAb anti-mouse/rat IL-1β (B122, BioXcell) for the indicated duration. Cells were stimulated with f(ab')$_2$ fragments of goat anti-mouse IgM antibody (eBioscience, San Diego, CA), recombinant mouse IL-6 (R&D Systems), recombinant mouse IL-21 (R&D Systems), recombinant mouse IL-10 (R&D Systems), recombinant mouse IL-1β (R&D Systems), CpG oligodeoxynucleotide 1826, or *Escherichia coli* lipopolysaccharide (Sigma Aldrich) for the indicated duration.

## Small interfering RNA (siRNA)

The following siRNAs were used in this study: Accell Mouse Stat5a (20850) siRNA-SMART pool (Dharmacon, Cambridge, UK) and Accell Non-targeting Pool (Dharmacon). Transfection was performed by following the manufacture's instruction. Briefly, isolated cells were incubated with 1 µM Accell siRNA in Accell siRNA Delivery Media (Dharmacon) at 37 °C, 5% $CO_2$ for 48 hr. Knockdown was assessed by Western blotting.

## RNA sequencing (RNA-seq)

Total RNA was extracted from isolated cells using the RNeasy Plus Mini kit (Qiagen, Germantown, MD) according to the manufacturer's instruction. Briefly, isolated cells were lysed in RLT Plus buffer. Prior to RNA isolation, genomic DNA (gDNA) was removed using a gDNA eliminator. RNA was extracted after washing with RW1 buffer and RPE buffer. RNA samples were processed following the protocol for the Clontech (Mountain View, CA) SMART-Seq V4 Ultra Low Input RNA Preparation Kit (first-strand cDNA synthesis; full-length double strand cDNA amplification by LD-PCR; amplified cDNA purification and validation). The cDNAs were then further prepared using Illumina (San Diego, CA) Nextera DNA Library Preparation Kit (tagmentation; PCR amplification; clean-up and validation). Paired-end sequencing (100x2 cycles paired end reads) of multiplexed mRNA samples was carried out on an Illumina HiSeq 2500 sequencer. Fastq files obtained from the sequencer were generated using FDA HIVE platform v2.4 (*Simonyan et al., 2016*). Quality of sequence reads was inspected using HIVE's native multi-QC tool. Sequence reads were mapped to GRCm38.p6 transcriptome (G CF_000001635.26) using HIVE's native NGS aligner Hexagon (v2.4) (*Santana-Quintero et al., 2014*). Gene-level feature counts were quantified using HIVE's native feature mapping tool, HIVE Alignment Comparator (v2.4) and normalized to RPKM (*Mascia et al., 2022*). Sequencing data are available in the GEO repository (Accession number GSE200955).

## Western blot analysis

Cultured cells were rinsed with ice-cold PBS with protease/phosphatase inhibitors (Thermo Fisher Scientific). Total cell lysates were prepared using RIPA buffer (Thermo Fisher Scientific) and protease/phosphatase inhibitors. The cell lysates were added into 4 x Laemmli loading buffer (Bio Rad, Hercules, CA) and denatured at 95 °C for 10 min. Protein samples were separated by tris-glycine denaturing SDS-PAGE (Bio Rad) and transferred onto nitrocellulose membranes (iBlot2, Invitrogen, Waltham, MA). Membranes were blocked with 5% bovine serum albumin or nonfat dry milk for 30 min, followed by incubation with primary antibodies overnight at 4 °C and HRP-conjugated secondary antibodies for 1 hr. The following antibodies used in blotting were from Cell Signaling Technology (Danvers, MA): anti-phospho STAT1 Tyr[701] (58D6), anti-STAT1 (42H3), anti-phospho STAT3

Tyr[706] (D3A7), anti-STAT3 (79D7), anti-phospho STAT5 Tyr[694] (D47E7), anti-STAT5 (D2O6Y), anti-beta-actin (D6A8), anti-phospho PKC (pan) betaII Ser[660], anti-phospho FAK Tyr[397], anti-FAK, anti-phospho Akt Ser[473] (D9E), anti-Akt (pan) (C67E7), anti-phospho p38 MAPK Thr[180]/Tyr[182] (D3F9), anti-p38 MAPK (D13E1), anti-phospho SAPK/JNK Thr[183]/Tyr[185] (81E11), anti-SAPK/JNK, anti-phospho p44/42 MAPK (Erk1/2) Thr[202]/Tyr[204] (D13.14.4E), anti-p44/42 MAPK (Erk1/2) (137F5), anti-phospho NF-κB p65 Ser[536] (93H1), anti-NF-κB p65 (D14E12), anti-phospho IκBα Ser[32] (14D4), anti-IκBα (44D4), anti-rabbit IgG, HRP-linked. Also, anti-PKC (A-3, Santa Cruz, Dallas TX), and anti-phospho FAK Tyr[576] (2H74L24, Invitrogen) were used in Western blot analysis. Acquired images were analyzed using ImageJ (NIH) software.

## Flow cytometry

For surface staining, cells were incubated with fluorochrome-labeled antibodies against surface proteins at 4 °C for 10 min, followed by 4',6-diamino-2-phenylindole (DAPI) staining during a 5 min centrifugation. For all intracellular staining, cells were incubated in PBS containing Zombie UV Fixable Viability Dye (BioLegend, San Diego, CA) at 4 °C for 20 min. Subsequently, for intracellular cytokine staining, cells were fixed/permeabilized with the Foxp3 staining buffer kit (Invitrogen) and then stained with fluorochrome-labeled antibodies against cytokines for 30 min at room temperature in Permeabilization Buffer (Invitrogen). For intracellular phosphorylated protein staining, cells were fixed with BD Cytofix (BD Biosciences) at 37 °C for 10 min and then permeabilized with ice-cold BD Phosflow Perm Buffer III (BD Biosciences) at 4 °C for 1 hr. Cells were stained with fluorochrome-labeled antibodies against phosphorylated proteins for 30 min at room temperature in PBS containing 0.05% fetal calf serum. The following fluorochrome-labeled antibodies were from BioLegend: phycoerythrin (PE)-anti-mouse CD43 (S11), Pacific Blue-anti-mouse CD19 (6D5), fluorescein isothiocyanate (FITC)-anti-mouse CD19 (1D3/CD19), allophycocyanin (APC)-Cy7-anti-mouse CD19 (6D5), Brilliant Violet (BV) 510-anti-mouse/human CD45R/B220 (RA3-6B2), FITC-anti-mouse IgM (RMM-1), PE-anti-mouse CD126 (IL-6Rα chain) (D7715A7), FITC-anti-mouse IL-10 (JES5-16E3), and PE-anti-mouse IL-6 (MP5-20F3). Also, BV605-anti-mouse IgD (11–26 c.2α) (BD Biosciences), FAK (Phospho-Tyr[397]) (CF405M) (biorbyt, Cambridge, UK), FAK (FITC) (biorbyt), anti-phospho STAT3 Tyr[706] (D3A7, Cell Signaling Technology), anti-phospho STAT5 Tyr[694] (D47E7, Cell Signaling Technology), APC-anti-mouse CD130 (gp130) (KGP130, Invitrogen), and Alexa Fluor 488-Donkey anti-Rabbit IgG (H+L) (Jackson ImmunoResearch Inc) were used in flow cytometry. Live cells were determined by forward Scatter (FSC), side scatter (SSC), DAPI staining or Zombie UV Fixable Viability Kit (BioLegend). Flow cytometric analysis of isolated B lymphocytes were performed using an LSR Fortessa flow cytometer (BD Biosciences) in the CBER Flow Cytometry Core Facility on the FDA White Oak campus (Silver Spring, MD). Data were analyzed using FlowJo software.

## Quantitative RT-PCR

To quantify gene expression, total RNA was extracted from isolated cells using the RNeasy Mini kit (Qiagen). Complementary DNA was synthesized from the extracted RNA using TaqMan Reverse Transcription Reagents (Applied Biosystems, Bedford, MA). Quantitative PCR was performed on CFX96 Touch Real-Time PCR Detection System (Bio Rad) using TaqMan Gene Expression Master Mix (Applied Biosystems) and TaqMan probes (Thermo Fisher Scientific). The following primers were used: *Gapdh* (Mm99999915_g1), *Il10* (Mm01288386_m1), *Il6* (Mm00446190_m1), *Il6ra* (Mm01211445_m1), *Tnf* (Mm00443258_m1), and *Il1b* (Mm00434228_m1). Gene expression was determined by the difference between the quantification cycle (Cq) of the gene of interest and the Cq of the reference gene (*Gapdh*) of the same sample (ΔCq). The ΔCq for each replicate was exponentially transformed to the ΔCq expression by the formula $2^{-\Delta Cq}$.

## Enzyme-linked immunosorbent assay (ELISA)

The levels of cytokines in cell culture medium were determined with BD OptEIA Mouse IL-6 ELISA set (BD Biosciences, Franklin Lakes, NJ), ELISA MAX Standard Set Mouse IL-10 (BioLegend), Mouse TNF-α DuoSet ELISA (R&D Systems), and LEGEND MAX Mouse IL-35 Heterodimer ELISA Kit (BioLegend) following manufacturer's instructions.

## Statistical analyses

All statistical analyses were performed using Prism8 (GraphPad, San Diego, CA). Data were generally presented as mean ± standard deviation (s.d.). Two-tailed Student's $t$-test, one-way analysis of variance (ANOVA) with a Dunnett's multiple comparisons test and two-way ANOVA tests were performed to determine significance. $p < 0.05$ was considered statistically significant.

## Acknowledgements

We acknowledge the valuable technical support by FDA/CBER Veterinary Services and the Flow Cytometry Core Facility.

This study was conducted with the US FDA intramural funds. Also, this project was supported in part by an appointment to the Research Fellowship Program at the OVRR/CBER, U.S. Food and Drug Administration, administered by the Oak Ridge Institute for Science and Education through an interagency agreement between the U.S. Department of Energy and FDA.

## Additional information

### Funding
No external funding was received for this work.

### Author contributions
Jiro Sakai, Conceptualization, Data curation, Formal analysis, Methodology, Writing - original draft, Writing - review and editing; Jiyeon Yang, Investigation, Writing - review and editing; Chao-Kai Chou, Data curation, Investigation, Writing - review and editing; Wells W Wu, Data curation, Writing - review and editing; Mustafa Akkoyunlu, Conceptualization, Resources, Formal analysis, Supervision, Funding acquisition, Writing - original draft, Project administration, Writing - review and editing

### Author ORCIDs
Jiro Sakai (iD) http://orcid.org/0000-0002-2526-2766
Mustafa Akkoyunlu (iD) http://orcid.org/0000-0001-9958-4031

### Ethics
The mouse experiments described in this study were performed in accordance with the US Food and Drug Administration/Center for Biologics Evaluation and Research Institutional Animal Care and Use Committee guidelines (permit 2002-31 and 2017-45).

### Decision letter and Author response
Decision letter https://doi.org/10.7554/eLife.83561.sa1
Author response https://doi.org/10.7554/eLife.83561.sa2

## Additional files

### Supplementary files
• MDAR checklist

### Data availability
Sequencing data have been deposited in GEO under accession code GSE200955.

The following dataset was generated:

| Author(s) | Year | Dataset title | Dataset URL | Database and Identifier |
|---|---|---|---|---|
| Sakai J, Chou C, Akkoyunlu M | 2022 | Next Generation Sequencing analysis of BCR-induced gene expression in adult and neonatal splenic B cells | https://www.ncbi.nlm.nih.gov/geo/query/acc.cgi?acc=GSE200955 | NCBI Gene Expression Omnibus, GSE200955 |

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
