## [Editor Report]

In this study, authors demonstrate that neonatal mice produce more CD43- B cell-derived IL-10 following anti-BCR stimulation than adult mice. This is due to a autocrine mechanisms where by anti-BCR stimulation leads to pSTAT5 up regulation, production of IL-6 which then enhances IL-10 production via pSTAT3. These are interesting results for the regulatory B cell field, demonstrating that signaling is different in adult vs neonatal B cells and in particular for researchers studying the mechanisms underpinning the enhanced susceptibility to infection.

---

## [Decision Letter]

**Decision letter after peer review:**

Thank you for submitting your article "B cell receptor induced IL-10 production from neonatal CD19^+^CD43-cells depends on STAT5 mediated IL-6 secretion" for consideration by *eLife*. Your article has been reviewed by 2 peer reviewers, including Tomoharu Yasuda as Reviewing Editor and Reviewer #2, and the evaluation has been overseen by Betty Diamond as the Senior Editor. The following individual involved in the review of your submission has agreed to reveal their identity: Lizzy Rosser (Reviewer #1).

Essential revisions:

Experimentally,

1) Characterization of the neonatal CD43-negative B cell population has not been carefully performed. It should be useful if authors show the expression of surface markers for B10, B1, and B2 cells such as CD1d, CD5, B220, CD21, CD23, and CD93.

2) Figure 1B: Because CD43 is one of the activation markers for B cells, CD43 levels could be changed in the course of stimulation. Therefore, the authors should stimulate and analyze IL-10 levels in B cells after the separation of CD19^+^CD43- and CD19^+^CD43+ fractions.

3) It is not convincing whether the MACS-depleted CD43 negative B cell fraction used for entire experiments contains only CD43-negative B cells or includes CD43-low B cells. If the CD43 expression level is similar to adult splenic B2 cells, it is not the matter. But if it contains CD43low cells, authors should describe them as CD43-/low cells rather than CD43- cells. This point may be addressed by comparing RNAseq data used in Figure 2A or MFI comparison between fractions from adults and neonates.

4) Mature splenic B cells in adult mice downregulate IgM while upregulating IgD. Different IgM expression levels in neonatal CD19^+^CD43+ B cells and adult CD19^+^CD43+ B cells could explain the different reactivity to BCR crosslinking. FACS analysis of IgM/IgD levels in those cells will be important to interpret the data.

5) IL10 staining is marginal and not convincing. The addition of brefeldin/monensin in combination with BCR stimulation should be tried if the authors have not done yet.

6) According to reviewer #1's suggestions, the text should be carefully edited; particularly, in the introduction and discussion, the authors should fairly refer to other peoples' data for non-expert readers to easily understand.

*Reviewer #1 (Recommendations for the authors):*

Suggestions for the authors:

Introduction:

1 – Please refer to CD43- as B-2 cells not non B-1 cells.

2 – CD5^+^CD1dhi B cells are usually referred to as B10 cells, whilst regulatory B cells refer to all IL-10-producing B cells.

3 – Multiple cytokines further to IL-1b and IL-6 have now been shown to induce IL-10 production by B cells, most notably IFN. This is important as previous research for context has been type 1 interferon protects neonatal from acute inflammation via induction of IL-10-producing B cells (Zhang et al. 2007, JEM).

4 – The authors should be clear in the introduction about what sites they are talking about (i.e. the spleen) and when data is referencing human or murine studies.

Results:

1 – It would be good to have more explanation about the use of anti-BCR stimulation in vitro alone rather than the addition of other agents such as anti-CD40 in combination or TLR agonists. These agents are more well known to stimulate IL-10 production rather than solo stimulation with anti-BCR.

2 – In figure 1, the seems to be only a marginal shift to suggest positive IL-10 staining. It is unclear from the methods whether the authors added brefeldin/monensin prior to staining intracellular antigens. Is there a rationale for using Fix/Perm intra-nuclear fixation buffer rather than a cytoplasmic fixation buffer?

3 – Figure 2 supplementary data (especially the number of DEGs) should be in the main figure, could the authors improve the presentation of the pathway data?

4 – I thought that the work using different inhibitors to assess differences in pathway activation between adults and neonates was well executed.

5 – Figure 5 – The IL-6ra staining is not that convincing, perhaps a matched PCR for the gene would be useful? And the error bar is very wide, so higher numbers would be informative.

6 – Why do the authors think that IL-6 does not induce IL-10 from B-1 cells? Is this due to differences in pSTA5/pSTAT3 signalling downstream of B-1 versus B-2 BCR?

7 – Is the signaling cascade downstream of IL-6R signaling also different in neonatal versus adult B cells?

Discussion:

1 – Why do the authors think that neonatal B cells show heightened BCR activation compared to adult B cells? Not experimentally to be addressed, but I would be interested to hear in the discussion what the authors thought the in vivo implications of these data were.

2 – How do the authors integrate their data with other studies showing different signals (e.g. IFN) that control the up-regulation of IL-10 by neonatal B cells?

Additional comments:

1 – What is the sex of the neonatal donor cells used? As female cells were used for adult B cells.

2 – Are triplicates in represented experiments biological or technical replicates? Numbers in general are very low for an in vitro study so it would be better to show combined data rather than representative data and as individual dots rather than error bars.

3 – Could the authors change the PCR axis to say δ CT as this is what they are presenting on the graphs?

*Reviewer #2 (Recommendations for the authors):*

Overall, this is a nice study identifying a unique signaling pathway leading to IL-10 production from neonatal B cell subset after the BCR stimulation. An interesting IL-10-producing mechanism was proposed based on solid and ample data. However, the neonatal CD43- B cell subset secreting IL-10 has not been well characterized and carefully analyzed. Thomas Tedder's group previously identified the CD19^+^IL-10+ B cell population in neonates by showing beautiful intracellular IL-10 staining. Those cells were CD1dlowCD5^+^ phenotype but may not be documented CD43 expression levels while adult B10 cells were divided in CD43+ and CD43- fractions. My major recommendation to improve this study would be to add careful analysis in the neonatal CD43- B cell subset secreting IL-10.

1) Characterization of the neonatal CD43-negative B cell population has not been carefully performed. It would be useful if authors show the expression of surface markers for B10, B1, and B2 cells such as CD1d, CD5, B220, CD21, CD23, and CD93.

2) Figure 1A and 1B: Intracellular staining of IL-10 is not convincing. I wonder if adding monensin in the culture medium during the anti-IgM stimulation improves intracellular IL-10 staining.

3) Figure 1B: Because CD43 is one of the activation markers for B cells, CD43 levels could be changed in the course of stimulation. Therefore, the authors should stimulate and analyze IL-10 levels in B cells after the separation of CD19^+^CD43- and CD19^+^CD43+ fractions.

4) It is not convincing whether the MACS-depleted CD43 negative B cell fraction used for entire experiments contains only CD43-negative B cells or includes CD43-low B cells. If the CD43 expression level is similar to adult splenic B2 cells, it is not the matter. But if it contains CD43low cells, authors should describe them as CD43-/low cells rather than CD43- cells. This point may be addressed by comparing RNAseq data used in Figure 2A or MFI comparison between fractions from adults and neonates.

5) Mature splenic B cells in adult mice downregulate IgM while upregulating IgD. Different IgM expression levels in neonatal CD19^+^CD43+ B cells and adult CD19^+^CD43+ B cells could explain the different reactivity to BCR crosslinking. FACS analysis of IgM/IgD levels in those cells will be important to interpret the data.

---

## [Author Response]

Essential revisions:Experimentally,1) Characterization of the neonatal CD43-negative B cell population has not been carefully performed. It should be useful if authors show the expression of surface markers for B10, B1, and B2 cells such as CD1d, CD5, B220, CD21, CD23, and CD93.

We agree that further characterization of CD43-negative B cell population will strengthen the conclusions drawn from the rest of the experiments. We now performed additional flow cytometry experiments to analyze IL-10 producer CD43-negative B cells described in the literature. We showed that the frequencies of T2-Marginal zone precursor cells, MZP (CD19^+^CD21^hi^CD23^hi^CD24^hi^) and marginal zone B cells, MZB (CD19^+^CD21^hi^CD23^-^) were significantly lower in neonates than adults. Tim-1 B cell population was comparable between the age groups (Figure 1—figure supplement 7). In contrast to these CD43-negative subsets, there was significantly higher frequency of CD5^+^CD1d^+^ as well as CD5^+^CD1d^hi^ populations in neonates than the adults and these populations produced the highest levels of IL-10 (Figure 1E and Figure 1—figure supplement 8A).

2) Figure 1B: Because CD43 is one of the activation markers for B cells, CD43 levels could be changed in the course of stimulation. Therefore, the authors should stimulate and analyze IL-10 levels in B cells after the separation of CD19^+^CD43- and CD19^+^CD43+ fractions.

We are in agreement with the editor/reviewer that the analysis of CD43 levels after stimulation is critical since we are focusing on CD43-negative cells following the stimulation of cells. However, when we tried to obtain CD19^+^CD43- and CD19^+^CD43+ fractions separately by the 2-step magnetic bead isolations using Pan mouse B Cell Isolation kit (130-095-813, Miltenyi), and the mouse CD43 (Ly-48) Microbeads (130049-801, Miltenyi) kit, we were not able to get pure CD43- and CD43+ populations from neonatal spleen. The Pan B Cells Isolation Kit appears to be optimized for adult splenocytes because it successfully removed non-B cells from adult spleen, but it failed to deplete non-B cells from neonatal spleen. CD43 microbeads however, worked well for both adult and neonatal cells. We also attempted to isolate cells by flow cytometry but found out that sorted neonatal cells became blunted and did not respond to stimulation after sorting. Thus, we were not able to identify an appropriate method to obtain separate fractions of CD19^+^CD43- and CD19^+^CD43+ cells from neonatal splenocytes.

As an alternative to separation of CD19^+^CD43- and CD19^+^CD43+ fractions, we now measured changes in CD43 expression before and after stimulation. We found that before stimulation adult CD19^+^ population contained very small population of CD43+ cells which were approximately 1/5^th^ of the neonatal CD43+ cells (Figure 1—figure supplement 6). There was an increase in the CD43+ population in both the age groups after stimulation. To assess whether the changes in CD43 expression had an impact on the IL-10 producing CD43-negative population, we subjected the CD43-negative population to different stringency of gating and measured IL-10^+^ cells among the CD43negative populations. We found that neonatal IL-10 producing CD43-negative population increased with more stringent gating of CD43^-^ cells after stimulation while adult CD43^-^IL-10^+^ population did not change (Figure 1—figure supplement 6). These results rule out the possible impact of BCR-induced changes in CD43 expression in IL10 production from CD43- population.

3) It is not convincing whether the MACS-depleted CD43 negative B cell fraction used for entire experiments contains only CD43-negative B cells or includes CD43-low B cells. If the CD43 expression level is similar to adult splenic B2 cells, it is not the matter. But if it contains CD43low cells, authors should describe them as CD43-/low cells rather than CD43- cells. This point may be addressed by comparing RNAseq data used in Figure 2A or MFI comparison between fractions from adults and neonates.

We thank the editor/reviewer for this comment. We now measured the CD43 MFI and CD43 gene (*spn*) expression in isolated CD19^+^CD43- population as suggested. We did not find any difference in the MFI or the gene expression levels between the age groups. These new data are now included as Figure 1—figure supplement 4A and B.

4) Mature splenic B cells in adult mice downregulate IgM while upregulating IgD. Different IgM expression levels in neonatal CD19^+^CD43+ B cells and adult CD19^+^CD43+ B cells could explain the different reactivity to BCR crosslinking. FACS analysis of IgM/IgD levels in those cells will be important to interpret the data.

We thank the editor/reviewer for pointing out the importance of measuring IgM and IgD levels between the age groups. We found no difference in CD19^+^IgM^+^ frequency between the age groups (Figure 1—figure supplement 5A). However, when we analyzed the subsets, we measured significantly higher frequency and MFI of IgM^hi^ cells among neonatal CD19^+^CD43^-^ population compared to those of adult cells. There was no difference in the IgM expression among the CD19^+^CD43^+^ population between the age groups. In contrast, both neonatal CD19^+^CD43^-^ cells and CD43^+^ B-1 cells exhibited lower frequency and MFI of IgD^hi^ cells compared to their adult counterparts (Figure 1figure supplement 5B). These results suggest that higher expression of IgM among the CD19^+^CD43^-^ cells may be responsible for the enhanced STAT5 phosphorylation and IL6 production in anti-IgM stimulated neonatal cells.

5) IL10 staining is marginal and not convincing. The addition of brefeldin/monensin in combination with BCR stimulation should be tried if the authors have not done yet.

We agree that IL-10 staining is marginal and brefeldin/monensin can, in general, help in capturing intracellular cytokine production. However, we were not able to use brefeldin/monensin during the culturing of the cells because these reagents also block the secretion of IL-6, which is required for the expression and secretion of IL-10 in neonatal cells. Nevertheless, our ELISA and q-PCR data complement the flow cytometry results to show that IL-10 is indeed produced more from CD43-negative neonatal B cells compared to adult cells and this outcome is dependent on initial IL-6 production following BCR engagement.

6) According to reviewer #1's suggestions, the text should be carefully edited; particularly, in the introduction and discussion, the authors should fairly refer to other peoples' data for non-expert readers to easily understand.

We have now expanded the introduction section to include the type I IFN reference the reviewer indicated (Reference 22). Also, we clarified the reasons to focus on BCR stimulation induced IL-10 production as it is relevant to vaccine responses. We also included additional clarifications in our Discussion section to provide insight into enhanced sensitivity of neonatal cells to BCR induced IL-10 production.

Reviewer #1 (Recommendations for the authors):Suggestions for the authors:Introduction:1 – Please refer to CD43- as B-2 cells not non B-1 cells.

We thank the reviewer for the suggestion to refer the CD43- as B-2 cells and not non-B-1 cells. We chose to refer this population as non-B-1 cells because CD43- cells include

B-2 cells as well as marginal zone B cells and Breg cells according to Nicole Baumgarth (reference 16 in the first version and reference 23 in the current version). Therefore, we respectfully prefer to keep the “non B-1 cell” definition.

2 – CD5^+^CD1dhi B cells are usually referred to as B10 cells, whilst regulatory B cells refer to all IL-10-producing B cells.

We thank the reviewer for pointing out this distinction. We now changed the text to replace “CD5^+^CD1d^hi^ B cells” with “IL-10-producing regulatory B cells” in line 82. Please also note that we now included data showing that the main IL-10 producing neonatal CD19-CD43- population is the B10 cells (CD5^+^CD1d^hi^) (Figure 1F).

3 – Multiple cytokines further to IL-1b and IL-6 have now been shown to induce IL-10 production by B cells, most notably IFN. This is important as previous research for context has been type 1 interferon protects neonatal from acute inflammation via induction of IL-10-producing B cells (Zhang et al. 2007, JEM).

We thank the reviewer for pointing out the Zhang et al. study showing the production of IL-10 by type I IFN. We now included this study in the introduction section (line 76).

4 – The authors should be clear in the introduction about what sites they are talking about (i.e. the spleen) and when data is referencing human or murine studies.

We thank the reviewer for drawing our attention to clarifying the sites and the species we refer to in the introduction. We now updated the introduction to include this information.

Results:1 – It would be good to have more explanation about the use of anti-BCR stimulation in vitro alone rather than the addition of other agents such as anti-CD40 in combination or TLR agonists. These agents are more well known to stimulate IL-10 production rather than solo stimulation with anti-BCR.

We would like to point out that the main reason we studied the responses of B cells to BCR stimulation is because our main focus is to provide insight into possible mechanisms of weak neonatal immune response to vaccines. Since antigen recognition is an early and crucial step in the initiation of vaccine responses, we used anti-IgM to engage BCR. We now clarified this point in the introduction section (First paragraph of Introduction. Line 53).

2 – In figure 1, the seems to be only a marginal shift to suggest positive IL-10 staining. It is unclear from the methods whether the authors added brefeldin/monensin prior to staining intracellular antigens. Is there a rationale for using Fix/Perm intra-nuclear fixation buffer rather than a cytoplasmic fixation buffer?

We thank the reviewer for this comment. Please see our response to editor’s comment

5.

3 – Figure 2 supplementary data (especially the number of DEGs) should be in the main figure, could the authors improve the presentation of the pathway data?

We thank the reviewer for this comment. We now updated the presentation and moved it to main figure section as Figure 2A.

4 – I thought that the work using different inhibitors to assess differences in pathway activation between adults and neonates was well executed.

We thank the reviewer for this comment.

5 – Figure 5 – The IL-6ra staining is not that convincing, perhaps a matched PCR for the gene would be useful? And the error bar is very wide, so higher numbers would be informative.

Although the difference in the frequency of IL-6Ra expressing CD43- cells is statistically significantly higher in neonates vs adults, we agree with the reviewer that the IL-6Ra levels are modest. We now performed q-PCR to measure the expression levels of adult and neonatal IL-6Ra gene. Neonatal cells expressed higher levels of *il6ra* but this difference was not statistically significant. We included this data to Results section (Figure 5-supp figure 1). Nevertheless, we clearly see a difference in p-STAT3 levels as well as the production of IL-10 in response to IL-6 stimulation between adult and neonatal cells. Based on the modest difference between adult and neonatal IL-6Ra expression, we now updated the Discussion section (paragraph 3) to indicate that additional factors downstream of IL-6Ra may contribute to the differences in IL-6 responses between adults and neonates.

6 – Why do the authors think that IL-6 does not induce IL-10 from B-1 cells? Is this due to differences in pSTA5/pSTAT3 signalling downstream of B-1 versus B-2 BCR?

We thank the reviewer for this relevant question. At this point we do not know why B-2 cells do not secrete IL-10 in response to IL-6. We also do not know if B-2 cells signal in response to IL-6. It is even possible that B-2 cells do not express IL-6R. We will examine these possibilities as a continuation of our current study.

7 – Is the signaling cascade downstream of IL-6R signaling also different in neonatal versus adult B cells?

So far, we have only measured STAT3 phosphorylation in response to IL-6 between adult and neonatal CD19^+^CD43- cells (Figure 5B). We showed that adult cell p-STAT3 levels also increase in response to IL-6, although this increase is less and delayed compared to the increase in neonatal cells. Our comparison of IL-6Ra expression levels partially explains this difference because the neonatal IL-6Ra are moderately higher than the levels on adult cells. There are likely other differences downstream of IL-6Ra, which need to be investigated further.

Discussion:1 – Why do the authors think that neonatal B cells show heightened BCR activation compared to adult B cells? Not experimentally to be addressed, but I would be interested to hear in the discussion what the authors thought the in vivo implications of these data were.

We have now shown that neonatal CD19^+^CD43^-^ B cells express higher levels of IgM (Figure 1—figure supplement 5) which may be amplifying the anti-IgM response. We also think that the elevated production of IL-10 in response to IgM stimulation may be needed to dampen the microbial inflammatory response as a result of the rapid encounter with microbes after birth. We now included this viewpoint in the last paragraph of Discussion section.

2 – How do the authors integrate their data with other studies showing different signals (e.g. IFN) that control the up-regulation of IL-10 by neonatal B cells?

We have ruled out the contribution of IL-1b in the production of IL-10 from CD19^+^CD43^-^

B cells but we did not specifically investigate the role for type I IFNs in BCR induced IL10 secretion. We believe that type I IFNs are unlikely to contribute to IL-10 secretion from CD19^+^CD43^-^ B cells because our anti-IL-6 blocking antibody experiments together with the IL-6 KO mouse B cell experiments clearly show that IL-10 production from antiIgM stimulated neonatal CD19^+^CD43^-^ B cells is mostly dependent on IL-6.

Additional comments:1 – What is the sex of the neonatal donor cells used? As female cells were used for adult B cells.

It is not possible to visually determine the sex of neonatal mice. We used all pups in our experiments without determining the sex of the neonatal mice. We included this information in the Materials and methods section (Cell isolation and culture).

2 – Are triplicates in represented experiments biological or technical replicates? Numbers in general are very low for an in vitro study so it would be better to show combined data rather than representative data and as individual dots rather than error bars.

The triplicates were biological replicates. We now included this information in each figure legend. We also changed the graphs to include individual data points.

3 – Could the authors change the PCR axis to say δ CT as this is what they are presenting on the graphs?

We thank the reviewer for this improvement suggestion. We now changed the PCR axis accordingly.

Reviewer #2 (Recommendations for the authors):Overall, this is a nice study identifying a unique signaling pathway leading to IL-10 production from neonatal B cell subset after the BCR stimulation. An interesting IL-10-producing mechanism was proposed based on solid and ample data. However, the neonatal CD43- B cell subset secreting IL-10 has not been well characterized and carefully analyzed. Thomas Tedder's group previously identified the CD19^+^IL-10+ B cell population in neonates by showing beautiful intracellular IL-10 staining. Those cells were CD1dlowCD5^+^ phenotype but may not be documented CD43 expression levels while adult B10 cells were divided in CD43+ and CD43- fractions. My major recommendation to improve this study would be to add careful analysis in the neonatal CD43- B cell subset secreting IL-10.

We thank the reviewer for recommending the characterization of the IL-10 producing CD43- B cell subsets. We have now performed this experiment. Please refer to our response to editor’s comment 2.

1) Characterization of the neonatal CD43-negative B cell population has not been carefully performed. It would be useful if authors show the expression of surface markers for B10, B1, and B2 cells such as CD1d, CD5, B220, CD21, CD23, and CD93.

We thank the reviewer for recommending the characterization of the IL-10 producing CD43- B cell subsets. We have now performed this experiment. Please refer to our response to editor’s comment 2.

2) Figure 1A and 1B: Intracellular staining of IL-10 is not convincing. I wonder if adding monensin in the culture medium during the anti-IgM stimulation improves intracellular IL-10 staining.

We thank the reviewer for this comment. In general, we agree that the intracellular IL-10 staining is not ideal. This is the same question as the editor’s comment 5. Please refer to our response to editor’s comment 5 for this question.

3) Figure 1B: Because CD43 is one of the activation markers for B cells, CD43 levels could be changed in the course of stimulation. Therefore, the authors should stimulate and analyze IL-10 levels in B cells after the separation of CD19^+^CD43- and CD19^+^CD43+ fractions.

We thank the reviewer for this comment. This is the same question as the editor’s comment 2. Please refer to our response to editor’s comment 2 for this question.

4) It is not convincing whether the MACS-depleted CD43 negative B cell fraction used for entire experiments contains only CD43-negative B cells or includes CD43-low B cells. If the CD43 expression level is similar to adult splenic B2 cells, it is not the matter. But if it contains CD43low cells, authors should describe them as CD43-/low cells rather than CD43- cells. This point may be addressed by comparing RNAseq data used in Figure 2A or MFI comparison between fractions from adults and neonates.

We thank the reviewer for this comment. This is the same question as the editor’s comment 3. Please refer to our response to editor’s comment 3 for this question.

5) Mature splenic B cells in adult mice downregulate IgM while upregulating IgD. Different IgM expression levels in neonatal CD19^+^CD43+ B cells and adult CD19^+^CD43+ B cells could explain the different reactivity to BCR crosslinking. FACS analysis of IgM/IgD levels in those cells will be important to interpret the data.

We thank the reviewer for this comment. This is the same question as the editor’s comment 4. Please refer to our response to editor’s comment 4 for this question.